# Caspase-1-licensed pyroptosis drives dsRNA-mediated necroptosis and dampens host defense against bacterial pneumonia

Qinyu Luo[1], Lihua Shen[1], Shiyue Yang[2], Yan Zhang[1], Yihang Pan[1], Zehua Wu[1], Qiang Shu[1]*, Qixing Chen [1]*

**1** Department of Clinical Research Center, The Children's Hospital, Zhejiang University School of Medicine, Hangzhou, China, **2** Department of Anesthesiology, The First Affiliated Hospital of Soochow University, Suzhou, China

* shuqiang@zju.edu.cn (QS); qixingchen@zju.edu.cn (QC)

## Abstract

Bacterial lung infections cause severe host responses. Here, we showed that global deficiency of caspase-1 can protect against lethal pulmonary *Escherichia coli* infection by reducing the necroptosis of infiltrated neutrophils, which are key players in immune responses in the lung. Mechanistically, neutrophil necroptosis was not directly triggered in a cell-intrinsic manner by invading bacteria but was triggered by bacteria-stimulated pyroptotic epithelial cell supernatants *in vitro*. In validation experiments, chimeric mice with nonhematopoietic caspase-1 or GSDMD knockout were protected from lung *E. coli* infection and exhibited decreased neutrophil death. Nonhematopoietic pyroptosis facilitates the release of dsRNAs and contributes to neutrophil ZBP1-related necroptosis. Moreover, blocking dsRNA or depleting ZBP1 ameliorated the pathophysiological process of pulmonary *E. coli* infection. Overall, our results demonstrate a paradigm of communication between necroptosis and pyroptosis in different cell types in cooperation with microbes and hosts and suggest that therapeutic targeting of the pyroptosis or necroptosis pathway may prevent pulmonary bacterial infection.

## Author summary

Bacterial lung infection has caused many deaths globally, leading to a major economic burden on health systems. The emergence of drug-resistant bacteria has reduced the efficacy of widely used antibiotics. Bacterial invasion induces multiple host responses, including regulated cell death; these pathways participate in the pathological process of lung infection and could be potential targets in the treatment of disease. However, the precise mechanism underlying this type of regulated cell death remains unclear. Here, using a bacterial lung infection

**Data availability statement:** The authors confirm that all data underlying the findings are fully available without restriction. All relevant data are within the paper and its Supporting Information files.

**Funding:** This study is supported by a program from the National Natural Science Foundation of China (82370013 to Q.C.). The funders had no role in the study design, data collection and analysis, decision to publish, or preparation of the manuscript.

**Competing interests:** The authors have declared that no competing interests exist.

model combined with several genetically engineered mice and chimeric mice, we show that two types of regulated cell death, pyroptosis and necroptosis, contribute to the pathobiology of bacterial lung infections. Furthermore, pyroptotic cells promote neutrophil necroptosis through a double-stranded RNA-related pathway. We also found that intervention in the pyroptosis–necroptosis communication pathway ameliorated bacterial lung infection in a mouse model. Our study provides new insights into the triggering mechanisms of regulated cell death and suggests novel targets for the treatment of bacterial lung infection.

## Introduction

Lower respiratory tract infections, such as pneumonia, lead to global health issues and cause more deaths worldwide than other infectious diseases [1]. Moreover, these infections are the leading cause of death in children among all diseases [2]. In the USA, community-acquired pneumonia accounts for 1.5 million hospitalizations annually and has a mortality rate of 30.6% [3]. Bacteria are the most commonly identified pathogens in patients with pneumonia [4,5]. Bacterial pneumonia leads to substantial morbidity and mortality and results in a major clinical and economic burden on health care systems worldwide [6]. More importantly, the emergence of drug-resistant bacteria has reduced the efficacy of antibiotics [7]. Thus, understanding the molecular and cellular mechanisms involved in the pathogenesis of bacterial pneumonia is urgently needed for the development of novel clinical therapeutic strategies.

The mechanisms underlying bacterial pneumonia remain unclear. Recently, inflammasomes have been proposed to participate in the pathogenesis of bacterial pneumonia [8]. The inflammasome is a cytoplasmic multiprotein complex that is expressed primarily in immune cells and barrier epithelial cells. As a first-line response in host–pathogen interactions, inflammasomes detect various sterile or infectious stimuli within the cell and induce subsequent cellular responses. During this process, multiple cytosolic pattern recognition receptors (PRRs) are responsible for stimulus sensing and the assembly of their inflammasomes. For example, the nucleotide-binding oligomerization domain-like receptor thermal protein domain associated protein 3 (NLRP3) inflammasome senses pore-forming toxin-induced potassium efflux, the nucleotide-binding oligomerization domain-like receptor family caspase recruitment domain containing 4 (NLRC4) inflammasome senses bacterial flagellin, and the NLRP6 inflammasome senses lipoteichoic acid in gram-positive bacteria. Nevertheless, these canonical inflammasomes all assemble and recruit the same adaptor protein, apoptosis-associated speck-like protein containing a caspase recruitment domain (ASC), and the same effector protein, pro-caspase-1. Upon inflammasome activation, caspase-1 cleaves pro-interleukin (IL)-1β and gasdermin D (GSDMD), leading to inflammatory cytokine release and lytic cell death [9]. Many studies have investigated the potential roles of these canonical inflammasomes in pneumonia. However, the conclusions are inconsistent and complicated. Cohen et al. and Kebaier et al. reported that depletion of NLRP3 or inhibition of caspase-1

increased the bacterial killing ability of macrophages and conferred resistance to *Staphylococcus aureus* pneumonia in mice [10,11]. Cohen et al. also reported that depletion of NLRC4 or inhibition of caspase-1 alleviated *Pseudomonas aeruginosa* pneumonia in mice [12]. However, in the case of pulmonary *Streptococcus pneumoniae* infection, McNeela et al. and Witzenrath et al. reported that the depletion of canonical inflammasomes worsens the disease, leading to poorer bacterial clearance and vital signs [13,14]. These findings suggest that additional processes interact with canonical inflammasome activation in a pathogen-specific manner to further control the pathogenesis of bacterial pneumonia.

Recently, necroptosis has been found to be involved in the pathogenesis of bacterial pneumonia. Necroptosis is a type of regulated cell death involving necrosis that occurs during development, inflammation and disease in multiple cell types, including neurons, epithelial cells, hepatocytes and immune cells [15,16]. Unlike pyroptosis or apoptosis, which are regulated by proteins from the caspase family, necroptosis is regulated mainly by the receptor-interacting protein kinase (RIPK) family [17]. Upon activation, the Toll/IL-1 receptor domain-containing adaptor protein inducing interferon-β (TRIF), Z-DNA binding protein 1 (ZBP1), or RIPK1 hetero-oligomerizes with RIPK3 and induces the assembly of an amyloid-like structure of RIPK3 homo-oligomers. These homo-oligomers act as docking stations for the recruitment of the pseudokinase mixed-lineage kinase domain-like (MLKL), the pore-forming executioner of necroptosis [18]. Necroptosis plays an important role in both noninfectious and infectious diseases. With respect to bacterial pneumonia, Gonzalez-Juarbe et al. reported that the administration of an MLKL inhibitor reduced bacterial titers and increased macrophage numbers during pulmonary *Serratia marcescens* infection in mice [19]. Kitur et al. reported that treatment with the RIPK1 inhibitor NEC-1s or depletion of RIPK3 increased bacterial clearance and facilitated immune cell infiltration during pulmonary *S. aureus* infection in mice [20]. However, how and through which pathways necroptosis is triggered during bacterial pneumonia remain unclear. Moreover, necroptosis is associated with inflammasome activation in several ways. Lee et al. reported that upon infection with certain viruses or bacteria, MLKL can be activated by absent in melanoma 2 (AIM2) and caspase-1-associated protein complexes termed PANoptosomes [21]. Whether necroptosis communicates with other cell death processes, such as inflammasome-related pyroptosis, during bacterial pneumonia is unclear.

Since caspase-1 represents the core component of various inflammasomes and *Escherichia coli* is a typical gram-negative bacterium that activates caspase-1 by stimulating various inflammasomes [22–24], we used a model of pulmonary *E. coli* infection in mice globally deficient in caspase-1 to address the aforementioned questions. We showed that the activation of caspase-1 contributed to the pathophysiology of pulmonary *E. coli* infection and facilitated lung neutrophil necroptosis. We found that neutrophil necroptosis was triggered not directly by bacteria but also by pyroptotic nonhematopoietic cells during *E. coli* infection. We further demonstrated that double-stranded RNA (dsRNA) and ZBP1 mediated neutrophil necroptosis and exacerbated pulmonary *E. coli* infection. Our results provide a previously undescribed paradigm of communication between necroptosis and pyroptosis in different cell types in response to microbial infection.

## Results

### Lung neutrophils are critical for defense against *E. coli* pneumonia in *Casp1⁻ᐟ⁻* mice

We first investigated the role of canonical inflammasomes in an *E. coli* pneumonia model using caspase-1 knockout (*Casp1⁻ᐟ⁻*) mice and *Casp1⁺ᐟ⁺* mice. Compared with the *Casp1⁺ᐟ⁺* mice, the *Casp1⁻ᐟ⁻* mice presented an increased survival rate during the 72-hour observation period after *E. coli* challenge (Fig 1A), suggesting that the caspase-1 inflammasome participates in the pathogenesis of *E. coli* pneumonia. We then monitored vital pulmonary signs of infection in the *Casp1⁻ᐟ⁻* and *Casp1⁺ᐟ⁺* mice at 12 hours after *E. coli* challenge. Consistent with the survival data, the *Casp1⁻ᐟ⁻* mice maintained their body temperature and respiratory rate (S1A and S1B Fig) and did not exhibit severe tissue injury (Fig 1B). The *Casp1⁻ᐟ⁻* mice also presented a decreased protein concentration, indicating less injury to the alveolar capillary barrier and a lower bacterial load in whole lung tissue (S1C and S1D Fig). In addition, significantly lower levels of two major proinflammatory cytokines, IL-1β and TNF-α, were observed in the *Casp1⁻ᐟ⁻* mice (S1E and S1F Fig).

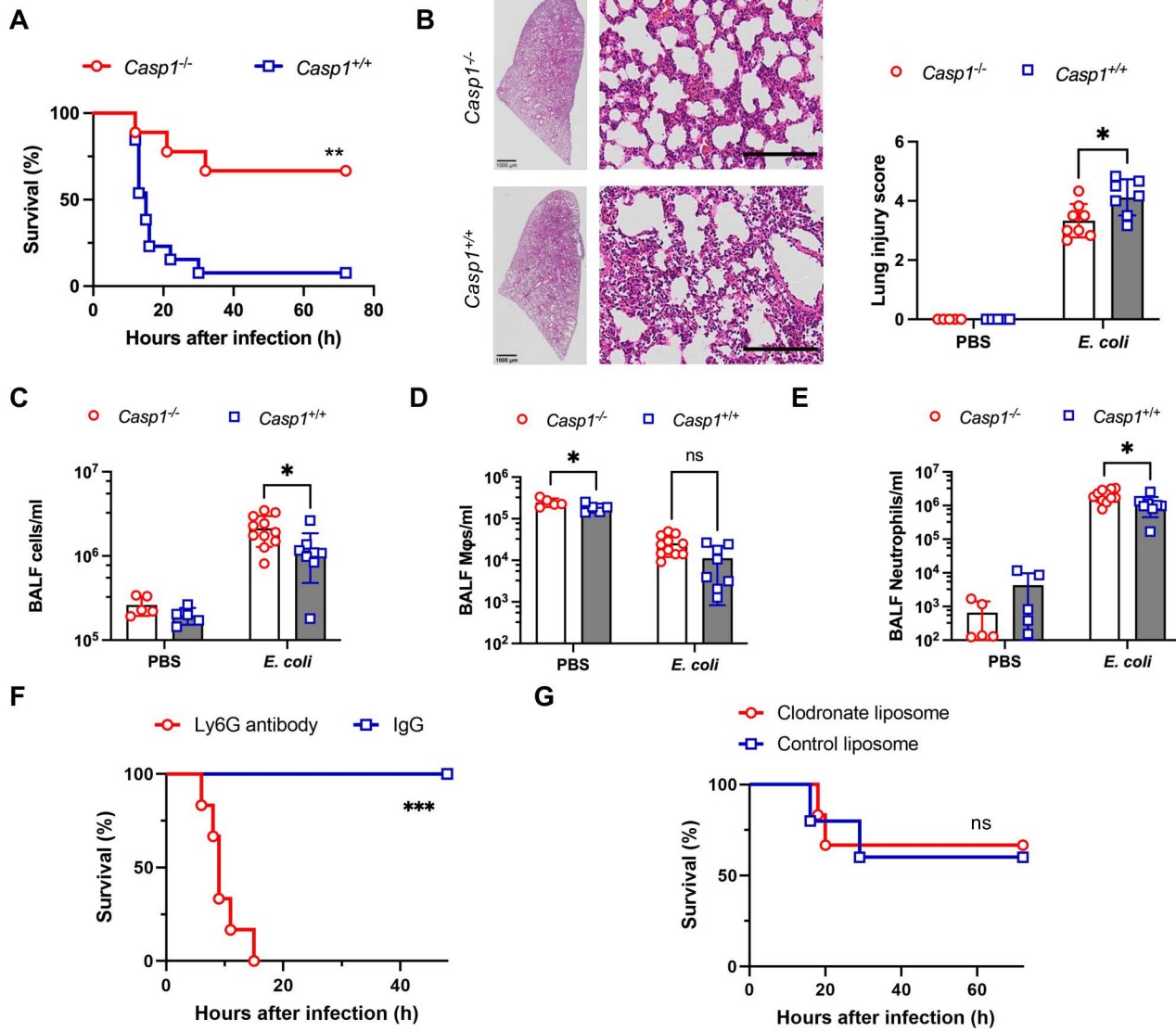

**Fig 1. Lung neutrophils are critical for defense against *E. coli* pneumonia in *Casp1-/-* mice. (A)** Survival of *Casp1-/-* mice (n=9) and *Casp1+/+* mice (n=13) after pulmonary *E. coli* infection. This experiment was conducted across three independent replicates, with each replicate including 3–5 mice. **(B)** Representative images of H&E-stained lungs from *Casp1-/-* mice (n=8) and *Casp1+/+* mice (n=7) at 12 hours after *E. coli* infection or intratracheal PBS instillation (n=5 each) and quantification of lung injury scores. Scale bars, 100 μm. **(C)** Counts of leukocytes in the BALF of *Casp1-/-* mice (n=11) and *Casp1+/+* mice (n=8) at 12 hours after *E. coli* infection or intratracheal PBS instillation (n=5 each). **(D and E)** Counts of macrophages and neutrophils in the BALF of *Casp1-/-* mice (n=11) and *Casp1+/+* mice (n=8) at 12 hours after *E. coli* infection or intratracheal PBS instillation (n=5 each). **(F)** Survival of *Casp1-/-* mice treated with anti-Ly6G antibody and IgG control antibody after pulmonary *E. coli* infection (n=6 each, conducted across two independent replicates, with each replicate including 3 mice). **(G)** Survival of *Casp1-/-* mice treated with clodronate liposomes (n=6) or control liposomes (n=5) after pulmonary *E. coli* infection. This experiment was conducted across two independent replicates, with each replicate including 2–3 mice. The data are shown as the means±SDs in (B, C, D and **E)**. Statistical differences were determined by the Mantel–Cox test (A, F and G) and two-way ANOVA (B, C, D and **E)**. *$P<0.05$; **$P<0.01$; ***$P<0.001$; ns, not significant. See also S1 Fig.

Leukocytes constitute the first-line immune defense against pathogen invasion, and they infiltrate infected tissue and perform immune functions. We then counted the number of leukocytes in the bronchoalveolar lavage fluid (BALF) at 12 hours after *E. coli* infection. We observed many more leukocytes in the *Casp1-/-* mice than in the *Casp1+/+* mice (Fig 1C).

We further used flow cytometry to characterize the immune cell composition of these leukocytes. While macrophages, especially SiglecF$^+$ tissue-resident macrophages, are the predominant cell type in the physiological state, neutrophils become more prominent at 12 hours after *E. coli* challenge (Fig 1D and 1E). Moreover, the number of neutrophils in the *Casp1*$^{-/-}$ mice was significantly greater than that in the *Casp1*$^{+/+}$ mice. Although we observed more tissue-resident macrophages in the *Casp1*$^{-/-}$ mice in the physiological state, these macrophages decreased dramatically at 12 hours post-*E. coli* infection. The number of recruited monocytes was comparable between the two groups after *E. coli* challenge (S1G and S1H Fig).

We also investigated whether the bactericidal function of neutrophils in *Casp1*$^{-/-}$ mice differed from that in *Casp1*$^{+/+}$ mice. We used flow cytometry to measure the production of reactive oxygen species (ROS), an essential procedure for intracellular bacterial killing, in neutrophils upon *E. coli* challenge. We found that the ROS levels in *Casp1*$^{-/-}$ neutrophils were comparable to those in *Casp1*$^{+/+}$ neutrophils after *E. coli* challenge (S1I Fig). Moreover, RNA sequencing (RNA-seq) analysis revealed that antibacterial defense signals were expressed at similar levels in *Casp1*$^{-/-}$ and *Casp1*$^{+/+}$ neutrophils (S1J Fig).

To further determine the role of neutrophils and macrophages in protecting against *E. coli*-induced pneumonia in *Casp1*$^{-/-}$ mice, we then used clodronate liposomes and an anti-Ly6G antibody to eliminate macrophages and neutrophils, respectively. Strikingly, the absence of neutrophils severely impaired host defense against *E. coli* infection in the *Casp1*$^{-/-}$ mice, whereas the deletion of macrophages had no effect (Fig 1F and 1G). These results indicate that neutrophils play a pivotal role in the pathogenesis of *E. coli* pneumonia in *Casp1*$^{-/-}$ mice.

### Decreased lung pyroptosis and infiltrating neutrophil necroptosis in *Casp1*$^{-/-}$ mice during pulmonary *E. coli* infection

We then investigated whether the difference in neutrophil number between *Casp1*$^{-/-}$ and *Casp1*$^{+/+}$ mice resulted from differential neutrophil chemotaxis in the experimental mice after *E. coli* challenge. We determined the neutrophil number in BALF at an earlier infection stage (4 hours after infection) and found no significant increase in the total cell count or neutrophil count in the *Casp1*$^{-/-}$ mice (S2A and S2B Fig). Additionally, at 12 hours after *E. coli* challenge, the numbers of peripheral white blood cells and neutrophils in the *Casp1*$^{-/-}$ and *Casp1*$^{+/+}$ mice were comparable (S2C and S2D Fig). We also determined the levels of the chemokines C-X-C motif ligand 1 (CXCL1) and CXCL2, both of which are important for neutrophil attraction during bacterial infection [25,26]. Unexpectedly, we found increased levels of both CXCL1 and CXCL2 in the BALF of the *Casp1*$^{+/+}$ mice (S2E and S2F Fig). These results indicate that the increased number of neutrophils in the lungs of *Casp1*$^{-/-}$ mice during *E. coli* infection is not dependent on chemotaxis.

We next investigated cell death, which may account for the difference in the number of neutrophils in the lungs of *Casp1*$^{-/-}$ and *Casp1*$^{+/+}$ mice after *E. coli* infection. We used propidium iodide (PI), a membrane-permeabilized fluorescent dye, to stain the nuclei of neutrophils undergoing lytic cell death. We found that, compared with the *Casp1*$^{+/+}$ mice, the *Casp1*$^{-/-}$ mice had a lower proportion of PI-stained neutrophils following instillation of *E. coli* (Fig 2A), suggesting that pulmonary neutrophils were preserved in a much more stable and living state in the *Casp1*$^{-/-}$ mice after *E. coli* challenge. A pathogenic bacterial strain is crucial in the study of human disease, as it closely mimics the pathogens responsible for human infection. Thus, we included a clinically isolated *E. coli* strain *in vivo*. We observed lower neutrophil death in the presence of this pathogenic *E. coli* (Fig 2B). We also measured lactate dehydrogenase (LDH) and high mobility group protein 1 (HMGB-1) levels in the BALF, both of which are released during lytic cell death, and observed significantly higher levels of LDH and HMGB-1 in the BALF of the *Casp1*$^{+/+}$ mice after *E. coli* infection (Figs 2C and S2G). However, these cell death-related substances can be released not only from dying neutrophils but also from other cells, such as lung parenchymal cells. Using terminal deoxynucleotidyl transferase-mediated dUTP nick-end labeling (TUNEL) staining, we detected many dead cells in the lung tissue of the *Casp1*$^{+/+}$ mice, which may consist of both parenchyma cells and immune cells (S3A Fig). We performed TUNEL staining for various cell markers in lung

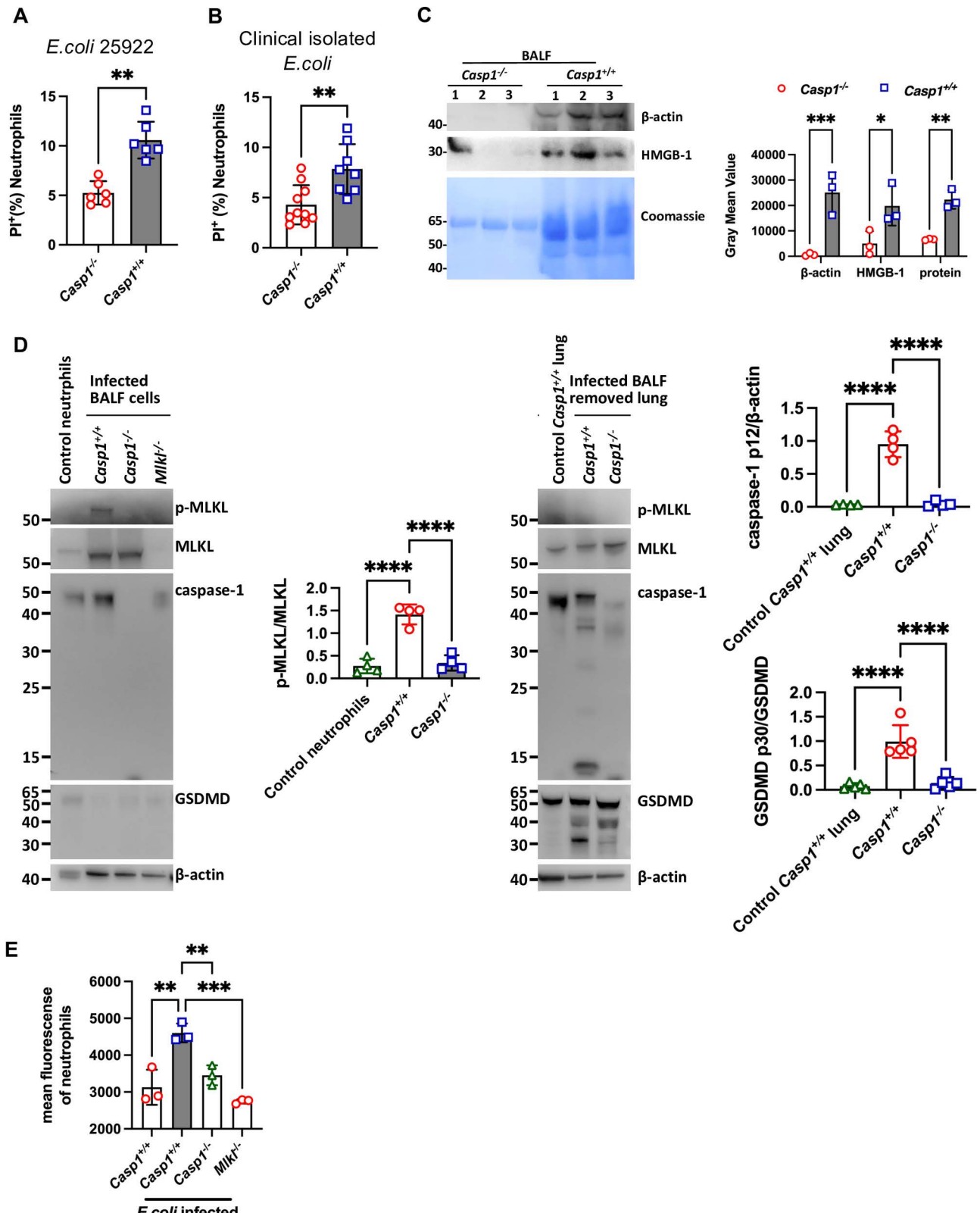

**Fig 2. Decreased lung pyroptosis and infiltrating neutrophil necroptosis in *Casp1*⁻ᐟ⁻ mice during pulmonary *E. coli* infection. (A and B)** Death of neutrophils in the BALF of *Casp1*⁻ᐟ⁻ mice and *Casp1*⁺ᐟ⁺ mice at 12 hours after infection with *E. coli* ATCC25922 (n = 6) or a clinically isolated *E. coli* strain

(n = 8-10). Cell viability was measured with PI staining and analyzed using flow cytometry. **(C)** HMGB1 levels in the BALF of *Casp1*<sup>-/-</sup> mice and *Casp1*<sup>+/+</sup> mice at 12 hours after *E. coli* infection. Each lane represents one mouse (n = 3). **(D)** BALF cells (containing > 95% neutrophils) and lung tissue without BALF cells from the indicated mice were collected and immunoblotted for programmed cell death-related proteins. **(E)** BALF neutrophils from the indicated mice were harvested at 12 hours after *E. coli* lung infection and stained with an anti-p-MLKL antibody, and the mean fluorescence was measured by flow cytometry. The data are shown as the means ± SDs in (A, B, C, D and E). Statistical differences were determined by Student's *t t*est (A, B and C) and one-way ANOVA (D and E). *$P < 0.05$; **$P < 0.01$; ***$P < 0.001$; ****$P < 0.0001$. See also S2, S3 and S4 Figs.

sections from the *Casp1*<sup>+/+</sup> and *Casp1*<sup>-/-</sup> mice (S3B Fig). Our findings indicated that both *Casp1*<sup>+/+</sup> and *Casp1*<sup>-/-</sup> mice presented comparable TUNEL signals in macrophages (F4/80<sup>+</sup>), with minimal signals observed in monocytes (CD14<sup>+</sup>) and type II epithelial cells (SPC<sup>+</sup>). Notably, the *Casp1*<sup>+/+</sup> mice presented significantly greater TUNEL signals in epithelial cells (CD326<sup>+</sup>), endothelial cells (CD31<sup>+</sup>), and neutrophils (Ly6G<sup>+</sup>), suggesting that type I epithelial cells, endothelial cells, and neutrophils are the primary cell types affected in *Casp1*<sup>+/+</sup> mice compared to *Casp1*<sup>-/-</sup> mice. These findings indicated that during pulmonary *E. coli* infection, both recruited neutrophils and lung parenchymal cells undergo certain types of programmed cell death. To verify this hypothesis, we then performed immunoblot analysis on cells collected from the BALF (containing >95% neutrophils) and remnant lung tissue after bronchoalveolar lavage to determine the related cell death modes using antibodies against the MLKL protein and GSDMD, which are hallmarks of necroptosis and pyroptosis [27,28]. We found that the BALF cells from *Casp1*<sup>+/+</sup> mice had substantially higher levels of phosphorylated MLKL (p-MLKL) than did the BALF cells from *Casp1*<sup>-/-</sup> mice, with a negative control from *Mlkl*<sup>-/-</sup> BALF cells (Fig 2D). In contrast, lung tissue from *Casp1*<sup>+/+</sup> mice presented elevated levels of cleaved GSDMD and activated caspase-1 (Fig 2D). Additionally, flow cytometry and immunofluorescence analysis confirmed that neutrophils from *Casp1*<sup>+/+</sup> mice presented higher levels of p-MLKL than did neutrophils from *Casp1*<sup>-/-</sup> mice (Figs 2E and S4A). We also assessed the location of p-MLKL in BALF-collected cells and found that it colocalized with PKH-26 (a probe indicating cell membranes; S4B Fig) on neutrophils (Ly6G<sup>+</sup>), which corresponds to its membrane pore-forming function and induction of neutrophil necroptosis. These findings further support the involvement of caspase-1 in neutrophil necroptosis during pulmonary *E. coli* infection.

## Intervention with MLKL decreased neutrophil death and protected mice against *E. coli* pneumonia

The aforementioned results suggest that via necroptotic death, neutrophils may participate in the pathogenesis of *E. coli*-induced pneumonia. Next, we intervened the core execution protein of necroptosis, MLKL, to verify these findings. We treated mice with an intraperitoneal injection of GW806742X [29] (an ATP mimetic that inhibits necroptosis by delaying MLKL membrane translocation) 1 hour before *E. coli* challenge. GW806742X administration significantly prolonged the survival period and increased the survival rate of wild-type (WT) mice (Fig 3A). However, we found no benefit for *Casp1*<sup>-/-</sup> mice after infection with an equal dose of *E. coli*, whereas a mild benefit was observed when they were infected with twice the dose of *E. coli* (Fig 3B). These findings suggest a potential alternative pathway independent of caspase-1 in regulating neutrophil necroptosis. Additionally, the administration of GW806742X markedly alleviated lung tissue injury, decreased the total protein concentration and bacterial load (S5A–S5C Fig), increased the total leukocyte number and neutrophil count and decreased BALF neutrophil death in WT mice (Fig 3C–3E), which mostly reproduced the phenomenon observed in infected *Casp1*<sup>-/-</sup> mice.

To further elucidate the role of MLKL in murine *E. coli* pneumonia, we utilized *Mlkl*<sup>-/-</sup> mice to confirm the role of neutrophil necroptosis in *E. coli*-induced pneumonia. Similarly, deletion of MLKL rescued the mice from lethal pneumonia, reduced overall lung injury, decreased the lung bacterial burden, increased neutrophil counts and decreased neutrophil death (Figs 3F–3H, S5D and S5E). These results indicated that inhibition of necroptosis can protect the mice against *E. coli*-induced lethal pneumonia, suggesting that neutrophil necroptosis underlies disease pathogenesis.

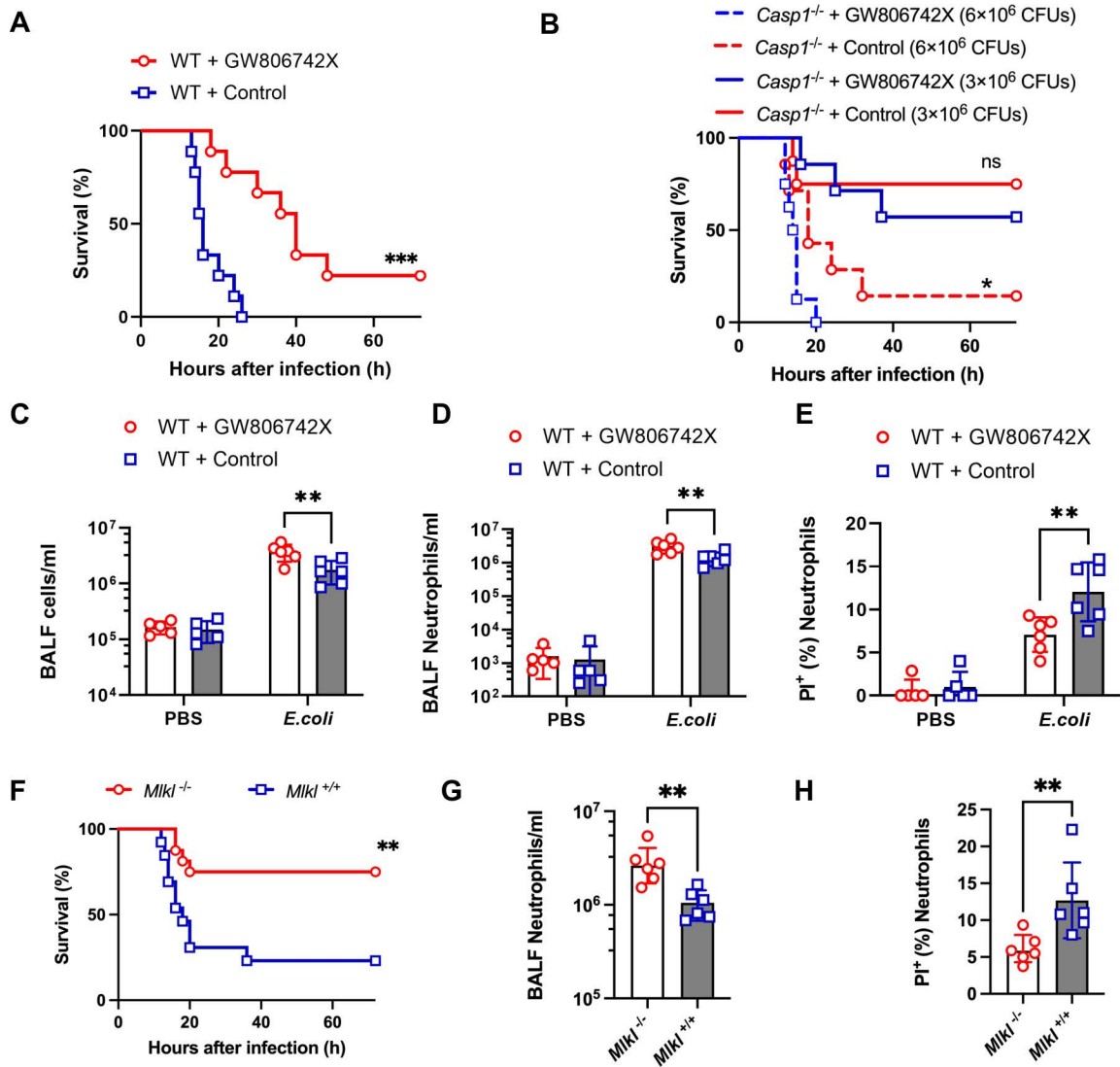

**Fig 3. Intervention with MLKL decreased neutrophil death and protected mice against *E. coli* pneumonia.** (A) Survival of WT mice treated with the MLKL inhibitor GW806742X or control solvent after pulmonary *E. coli* infection (n=9 each, conducted across three independent replicates, with each replicate including 3 mice). (B) Survival of *Casp1⁻/⁻* mice treated with the MLKL inhibitor GW806742X (n=7) or control solvent (n=8) after pulmonary *E. coli* infection with a single dose of *E. coli* (3×10⁶ CFUs), and survival of *Casp1⁻/⁻* mice treated with the MLKL inhibitor GW806742X (n=8) or control solvent (n=7) after pulmonary infection with a double dose of *E. coli* (6×10⁶ CFUs). This experiment was conducted across three independent replicates, with each replicate including 2–3 mice. (C-E) BALF from WT mice treated with the MLKL inhibitor GW806742X or control solvent was harvested at 12 hours after intratracheal PBS instillation (n=5 each) or pulmonary *E. coli* infection (n=6 each), and the total leukocyte count (C), neutrophil number (D) and PI⁺ neutrophil proportion (E) were determined. (F) Survival of *Mlkl⁻/⁻* mice (n=16) and *Mlkl⁺/⁺* mice (n=12) after pulmonary *E. coli* infection. This experiment was conducted across three independent replicates, with each replicate including 4–6 mice. (G-H) BALF from *Mlkl⁻/⁻* mice and *Mlkl⁺/⁺* mice was harvested at 12 hours after pulmonary *E. coli* infection, and the neutrophil number (H) and PI⁺ neutrophil proportion (I) were determined (n=6 each). The data are shown as the means±SDs in (C, D, E, G and H). Statistical differences were determined by the Mantel–Cox test (A, B and F), two-way ANOVA (C, D and E) and Student's *t* test (G and H). *$P<0.05$; **$P<0.01$; ***$P<0.001$. ns, not significant. See also S5 Fig.

## Nonhematopoietic pyroptosis contributes to neutrophil necroptosis

Recent studies have shown that certain pathogens can trigger the formation of the PANoptosome, a multiprotein complex. This complex is composed of caspase-1, caspase-8, RIPK1, RIPK3, Fas-associated protein with death domain,

apoptosis-related spot-like protein, AIM2, pyrin, and ZBP1. The PANoptosome induces PANoptosis, a process that includes apoptosis, pyroptosis, and necroptosis [21]. We next speculated that the observed neutrophil necroptosis is a cell-intrinsic mechanism associated with PANoptosis upon the sensing of certain external stimuli. To test this hypothesis, we used live *E. coli*, lipopolysaccharide (LPS) and heat-inactivated *E. coli* to stimulate neutrophils and mimic possible infection conditions *in vivo*. However, we did not find any significant difference in cell death between *Casp1*[+/+] and *Casp1*[-/-] neutrophils after stimulation (S6A Fig). Immunoblot analysis also revealed no difference in p-MLKL levels between *Casp1*[+/+] and *Casp1*[-/-] neutrophils (S6B Fig). These findings suggested that during *E. coli* infection, pulmonary neutrophil necroptosis is not caused by an intracellular mechanism associated with caspase-1-dependent PANoptosis.

Necroptosis can also potentially be triggered by inflammatory mediators such as TNF-α, IL-1β or HMGB-1. Among these mediators, IL-1β is the cytokine most relevant to caspase-1 because caspase-1 cleaves pro-IL-1β into its mature form. The activation of IL-1R, the receptor of IL-1β, reportedly limits necroptosis in certain kidney diseases [30]. TNF-α is associated with necroptosis since TNF-α triggers necroptosis when proapoptotic caspase-8 is inhibited [31]. HMGB-1 has also been reported to affect the triggering of necroptosis by pathogen toxins [32]. We then investigated whether the elevated levels of these mediators were responsible for neutrophil necroptosis during pulmonary *E. coli* infection. We used these mediators as stimuli for neutrophil necroptosis and found that none of the mediators could induce significant cell death regardless of their concentration (S6C Fig). We also administered the IL-1 receptor antagonist anakinra to competitively inhibit the function of IL-1β *in vivo* or, alternatively, used an anti-IL-1β antibody to neutralize IL-1β during *E. coli* infection. These results confirmed that blocking IL-1β did not mitigate the severity of lethal pneumonia in mice (S6D–S6F Fig). Thus, these results exclude the possible role of inflammatory mediators in neutrophil necroptosis during pulmonary *E. coli* infection.

Since western blot analysis of lung tissue revealed that caspase-1 and GSDMD were activated in *Casp1*[+/+] mouse lungs (Fig 2D), we further postulated that neutrophil necroptosis may be a consequence of lung parenchyma cell pyroptosis. We then incubated a single-cell suspension from the lung tissue of *E. coli*-infected WT mice with FLICA, a probe used to assess caspase-1 activity, to determine which type of cell is associated with highly active caspase-1. We found that among the cells in the lung, the cell type with the highest proportion of active caspase-1 (marked by FLICA[+]) was epithelial cells and endothelial cells (Figs 4A and S7A). These results indicated that these nonhematopoietic cells are inflammasome-activated cells during lung *E. coli* infection.

Inflammasomes in barrier epithelial cells can sense certain pathogens and then trigger pyroptosis. In this study, we chose MLE-12 murine lung epithelial cells as an *in vitro* model. MLE-12 cells were first incubated with live *E. coli,* and pyroptosis was assessed by detecting the activation of caspase-1 and GSDMD in the culture supernatant (S7B Fig). We defined the supernatant from live *E. coli*-stimulated MLE-12 cells as 'pyroptotic supernatant' and used it as a subsequent stimulus for neutrophils. In parallel, we also tested supernatant from freeze-thawed MLE-12 cell lysates, supernatant from cultured *E. coli*, and a mixture of these supernatants to stimulate neutrophils. A substantial amount of neutrophil death was induced by the pyroptotic supernatant from live *E. coli*-stimulated MLE-12 cells (Fig 4B), whereas neither the supernatant from the cell lysate nor that from cultured *E. coli* alone induced neutrophil death. Interestingly, although these are distinct stimuli, combining the two supernatants resulted in significant neutrophil death (Fig 4B). Notably, this pyroptotic supernatant also induced comparable cell death in *Casp1*[-/-] neutrophils (S7C Fig). Immunoblot analysis confirmed elevated p-MLKL levels and phosphorylated levels of RIPK3 (p-RIPK3) during the stimulation of neutrophils with pyroptotic supernatant (Fig 4C). Concurrently, increased p-RIPK3 levels were also detected in pulmonary BALF cells after *E. coli* infection (Fig 4D, including L929 cells treated with TNF-α and Z-VAD as a positive control of p-RIPK3).

We next utilized caspase-1 and GSDMD siRNAs to suppress their activity in the epithelial cell line MLE-12 during *E. coli* challenge *in vitro* and then collected the culture supernatant to stimulate neutrophils. As expected, the supernatants from the caspase-1- and GSDMD siRNA-treated MLE-12 cells were unable to induce neutrophil death (Fig 4E and 4F). These results suggest that during *E. coli* infection, pulmonary neutrophil necroptosis may not arise from an

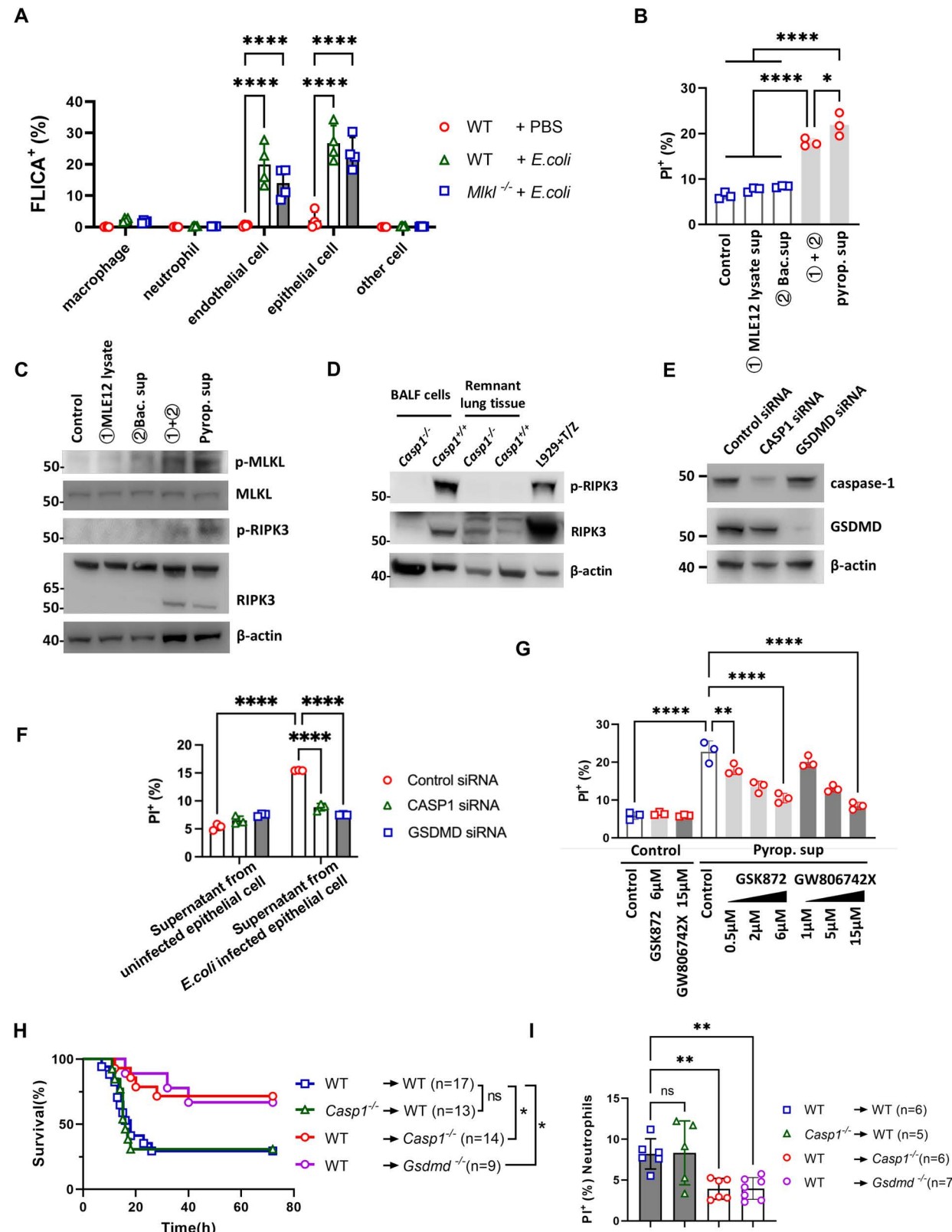

**Fig 4. Nonhematopoietic pyroptosis contributes to neutrophil necroptosis.** (A) The proportions of neutrophils, macrophages, endothelial cells, epithelial cells and other cells with activated caspase-1 in the lungs of WT and *Mlkl*⁻/⁻ mice at 12 hours after pulmonary *E. coli* infection were assessed

by FLICA staining using flow cytometry (n = 4). (B) Neutrophil death after stimulation with MLE12 lysate supernatant (MLE12 lysate sup), *E. coli* culture supernatant (bac. sup), a mixture of these supernatants, and pyroptotic supernatant (pyrop. sup) or control medium (n = 3). (C) Immunoblot of neutrophils stimulated with MLE12 cell lysate sup or bac. sup and a mixture of these supernatants and pyrop. sup as well as control medium. (D) BALF cells and lung tissue without BALF cells from *Casp1*[+/+] and *Casp1*[-/-] mice were collected 12 hours after *E. coli* infection and immunoblotted for RIPK3 and p-RIPK3. Lysate of L929 cell line treated with TNF-α and Z-VAD was used as a positive control for p-RIPK3. (E) Immunoblotting for caspase-1 and GSDMD in MLE-12 cells treated with caspase-1 or GSDMD siRNA. (F) Supernatants from PBS- or *E. coli*-stimulated MLE-12 cells pretreated with caspase-1 or GSDMD siRNA were collected and used to stimulate neutrophils, and cell death was evaluated by PI staining and flow cytometry (n = 3). (G) Neutrophils were pretreated with GSK872 and GW806742X at the indicated concentrations and then stimulated with pyrop. sup and control media, and cell death was assayed by using flow cytometry (n = 3). (H and I) Chimeric mice were generated via reciprocal bone marrow transplantation of WT, *Casp1*[-/-] and *Gsdmd*[-/-] mice as indicated, after which the mice were subjected to *E. coli*-induced pneumonia. Mouse survival was monitored (H). BALF cells were collected at 12 hours, and neutrophil death was determined by PI staining and flow cytometry (I). The survival experiment was conducted across three independent replicates, with each replicate including 3–6 mice. The data are shown as the means ± SDs in (A, B, F, G and I) and are representative of 3 independent experiments in (C, D, and E). Statistical differences were determined by two-way ANOVA (A, F and G), one-way ANOVA (B and I) or the Mantel–Cox test (H). *$P < 0.05$; **$P < 0.01$; ***$P < 0.001$; ****$P < 0.0001$. ns, not significant. See also S6 and S7 Figs.

intracellular mechanism associated with caspase-1 but may result from certain stimuli released from pathogen-induced caspase-1-mediated pyroptotic epithelial cells.

We further explored the possible mechanism involved in this model. Inhibitors, including NEC-1, GSK-872 and GW806742X, were used to inhibit the activity of the critical molecules RIPK1, RIPK3 and MLKL, respectively, in the necroptotic pathway upon stimulation with pyroptotic supernatant. Additionally, the pan-caspase inhibitor Z-VAD-FMK was used. Strikingly, cell death was markedly suppressed by GSK872 and GW806742X in both WT and *Casp1*[-/-] neutrophils but not in *Mlkl*[-/-] neutrophils (Figs 4G, S7C and S7D). However, the inhibition of NEC-1 or pan-caspases did not affect neutrophil necroptosis (S7E Fig). To validate these findings, we employed a chimeric mouse model comprising WT hematopoietic cells and *Casp1*[-/-] or *Gsdmd*[-/-] nonhematopoietic hosts in the context of *E. coli* pneumonia. We found that, compared with the mice with WT nonhematopoietic cells, the mice with *Casp1*[-/-] or *Gsdmd*[-/-] nonhematopoietic cells exhibited a greater survival rate and significantly lower neutrophil death rate (Fig 4H and 4I). Taken together, these data indicate that during *E. coli* infection, pulmonary neutrophil death is induced by nonhematopoietic pyroptotic cells such as epithelial cells.

### Pyroptosis-released dsRNAs induce neutrophil necroptosis via ZBP1 during pulmonary *E. coli* infection

During necroptosis, RIPK3 is phosphorylated and activated by proteins containing the RIP homotypic interaction motif (RHIM). These RHIM-containing proteins include TRIF and ZBP1. Additionally, RIPK1, another RHIM-containing protein, contributes to RIPK3 activation [33,34]. We then investigated whether TRIF and ZBP1 mediate neutrophil necroptosis upon *E. coli* infection. RNA-seq analysis revealed that ZBP1 transcripts were abundant in BALF neutrophils from both *Casp1*[+/+] and *Casp1*[-/-] mice following pulmonary *E. coli* infection, whereas TRIF transcripts were not detectable (Fig 5A). These results were further confirmed by immunoprecipitation (IP) analysis, which not only validated ZBP1 expression but also demonstrated that ZBP1 physically interacts with RIPK3 in BALF neutrophils. This interaction suggests that ZBP1 plays a direct role in mediating neutrophil necroptosis during pulmonary *E. coli* infection (Fig 5B). ZBP1 is a known nucleic acid sensor that can recognize both microbial and host-derived nucleic acids, including dsRNA [35]. To determine whether dsRNA is involved in neutrophil necroptosis, we performed IP analysis and found that ZBP1 binds dsRNA in BALF neutrophils upon *E. coli* infection (Fig 5C). We next investigated whether pyroptotic death of epithelial cells during *E. coli*-induced pneumonia contributes to the release of dsRNA. We found that supernatants from *E. coli*-stimulated MLE-12 cells contained abundant dsRNA (Fig 5D). Similarly, we found lower levels of dsRNA in the BALF of the *E. coli*-infected mice generated from WT hematopoietic cells from *Casp1*[-/-] or *Gsdmd*[-/-] hosts (Fig 5E). These results demonstrate that nonhematopoietic pyroptosis plays a critical role in the release of dsRNA.

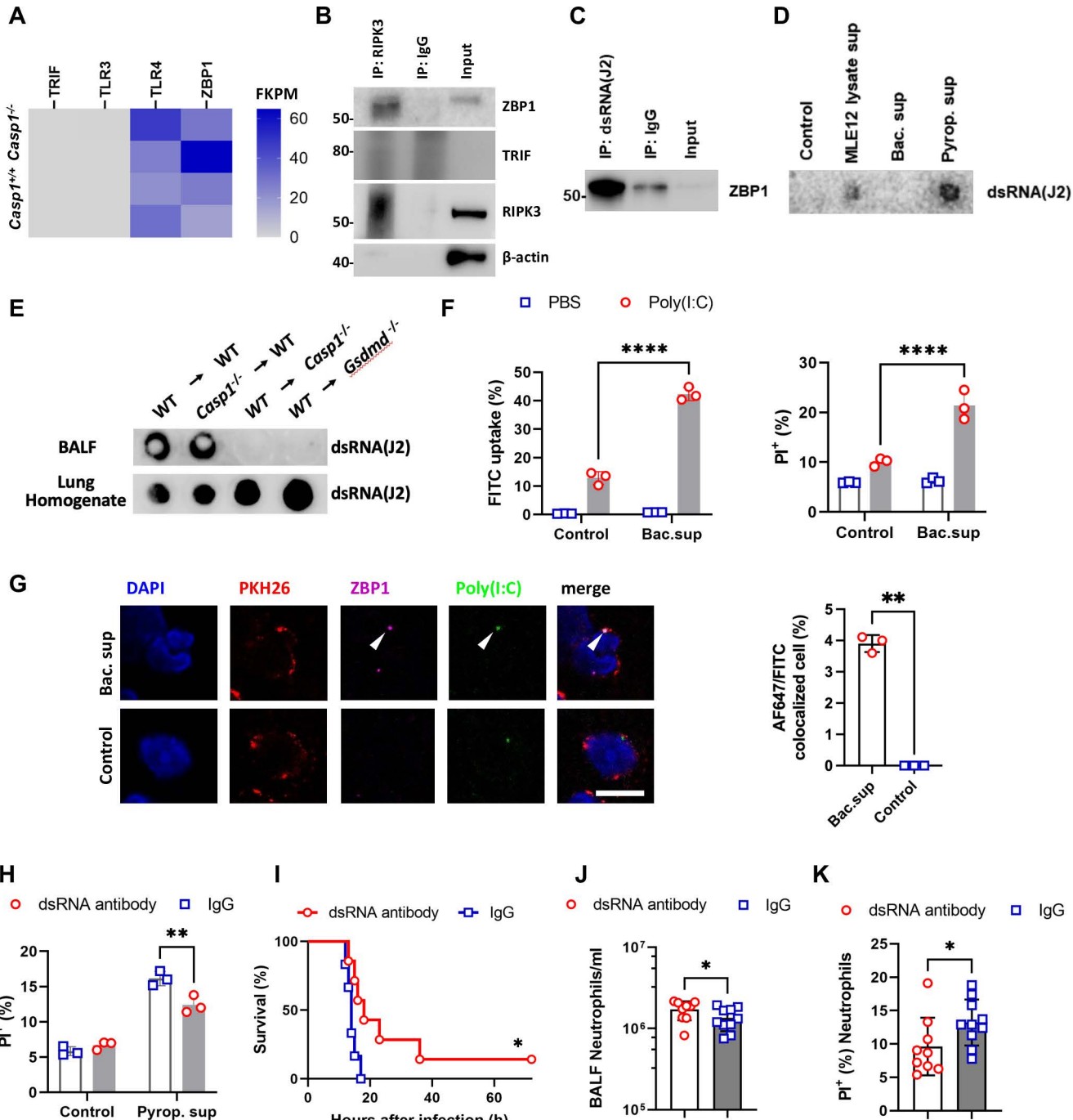

**Fig 5. Pyroptosis-released dsRNAs induce neutrophil necroptosis via ZBP1 during pulmonary *E. coli* infection.** (A) RNA sequencing of the gene expression of neutrophils in the BALF of *Casp1*[-/-] mice and *Casp1*[+/+] mice at 12 hours after pulmonary *E. coli* infection. FPKM, fragments per kilobase per million mapped fragments. n=2 for each group. (B) IP assay using anti-ZBP1 antibody or IgG control antibody in BALF cells from WT mice at 12 hours after pulmonary *E. coli* infection. (C) IP assay using anti-dsRNA antibody or IgG control antibody in BALF cells from WT mice at 12 hours after pulmonary *E. coli* infection. (D) Dot blot of MLE12 lysate sup, bac. Sup and pyrop. sup, as well as the control medium, via the dsRNA antibody. (E) Dot blot of BALF from the indicated chimeric mice at 12 hours after pulmonary *E. coli* infection using an anti-dsRNA antibody. (F) Neutrophil interactions of FITC-conjugated poly(I:C) and neutrophil death (PI[+]) were measured by flow cytometry after stimulation with *E. coli* culture supernatant (bac. sup) or control medium with/without FITC-conjugated poly(I:C) (n=3). (G) Representative immunofluorescence image of neutrophils cocultured with control/bac. sup and FITC-conjugated poly(I:C) using PKH26 and anti-ZBP1 antibodies and quantification of the percentage of AF647/FITC colocalized cells (*n*=3 biologically independent

samples). Scale bars, 10 µm. (H) Neutrophils were pretreated with an anti-dsRNA antibody or a control IgG antibody and stimulated with pyrop. sup or control medium, and cell death was assayed by using flow cytometry (n = 3). (I) Survival of mice pretreated intratracheally with anti-dsRNA antibody (n = 7) or IgG control antibody (n = 6) 1 hour before pulmonary *E. coli* infection. This experiment was conducted across three independent replicates, with each replicate including 2–3 mice. (J and K) Neutrophil numbers (J) and proportions of PI+ neutrophils (K) in the BALF of WT mice treated with anti-dsRNA antibody (n = 9) or IgG control antibody (n = 10) at 12 hours after pulmonary *E. coli* infection. The data are shown as the means ± SDs in (F, G, H, J and K) and are representative of 3 independent experiments in (B, C, D, E and G). Statistical differences were determined by the Mantel–Cox test (I), two-way ANOVA (F and H) and Student's *t* test (G, J and K). *$P < 0.05$; **$P < 0.01$; ****$P < 0.0001$. ns, not significant. See also S8 Fig.

To determine whether extracellular dsRNA can enter neutrophils, we cultured neutrophils with fluorescein isothiocyanate-labeled poly(I:C), a synthetic analog of dsRNA, in the presence of *E. coli* supernatant. Flow cytometric analysis revealed that poly(I:C) effectively interacted with neutrophils under these conditions and induced cell death (Fig 5F). Furthermore, immunocytochemical analysis revealed the colocalization of poly(I:C) with ZBP1 (Fig 5G), suggesting that dsRNA interacts with ZBP1 in neutrophils. To further assess the role of dsRNA in neutrophil death, we preincubated the pyroptotic supernatant with a dsRNA antibody before stimulating neutrophils. This treatment significantly reduced the ability of the pyroptotic supernatant to induce neutrophil death (Fig 5H). Moreover, administration of the dsRNA antibody significantly increased WT mice survival after *E.coli* infection (Fig 5I). Compared with the control antibody-treated mice, the mice treated with the dsRNA antibody also presented higher total cell counts and neutrophil numbers in the BALF (Figs 5J and S8A), a lower percentage of neutrophil death (Fig 5K), reduced protein leakage, increased bacterial clearance and decreased lung injury (S8B–S8D Fig). To study the redundancies between these pathways, we also treated *Casp1*$^{-/-}$, *Mlkl*$^{-/-}$ or *Zbp1*$^{-/-}$ mice with an anti-dsRNA antibody before *E. coli* pneumonia and found that dsRNA neutralization did not reduce neutrophil death in these mice after infection (S8E–S8G Fig). These results indicate that dsRNA neutralization ameliorates lung injury and promotes bacterial clearance by reducing neutrophil death and inflammation, but its protective effects depend on intact caspase-1-, MLKL-, and ZBP1-mediated pathways.

To confirm the role of ZBP1 in mediating neutrophil necroptosis *in vivo*, we conducted pulmonary *E. coli* infection in *Zbp1*$^{-/-}$ mice. Twelve hours post-infection, *Zbp1*$^{-/-}$ mice exhibited significantly reduced neutrophil death in the BALF (Fig 6A), along with increased BALF neutrophil counts and mild increases in total cell numbers (Fig 6B and 6C). Importantly, compared with their *Zbp1*$^{+/+}$ littermates, *Zbp1*$^{-/-}$ mice presented a significantly lower pulmonary bacterial burden and decreased lung injury (Figs 6D and S8H), similar to the effects observed in mice treated with the dsRNA antibody during pulmonary *E. coli* infection. We also used supernatants from *E. coli*-stimulated epithelial cells to validate the roles of ZBP1 and MLKL *in vitro*. As expected, significantly fewer *Zbp1*$^{-/-}$ and *Mlkl1*$^{-/-}$ neutrophils died after challenge with pyroptotic supernatants derived from epithelial cells (Fig 6E). TLR3 is a critical receptor for recognizing dsRNA, which plays a central role in initiating immune responses to viral infections. As shown by the RNA-seq data (Fig 5A) and western blot analysis (S9A Fig), the neutrophils from these mice did not express TLR3. We further used TLR3-deficient mice to conduct *in vivo* experiments. We found that TLR3 deficiency did not prevent neutrophil death or decrease the bacterial burden (S9B–S9E Fig). These results suggest that TLR3 may not be involved in this neutrophil necroptosis pathway. LPS is actively secreted during bacterial growth, contributing to the pathogenicity of *E. coli* infections. To investigate the role of LPS in this model, we treated purified neutrophils with LPS (1 µg/ml) and poly(I:C) (1 µg/ml). We found that this combination could trigger neutrophil death (Fig 6F). However, treatment with poly(I:C) in combination with bacterial supernatant resulted in significantly greater neutrophil death (Fig 5F). Thus, we believe that in addition to LPS, other components facilitate dsRNA-related cell death under such circumstances. Taken together, these data demonstrate that dsRNA can be released from *E. coli*-stimulated pyroptotic nonhematopoietic cells, such as epithelial cells, and interact with neutrophil ZBP1 to facilitate cell death. Furthermore, ZBP1 knockout or *in vivo* neutralization of dsRNA alleviated *E. coli* pneumonia in mice.

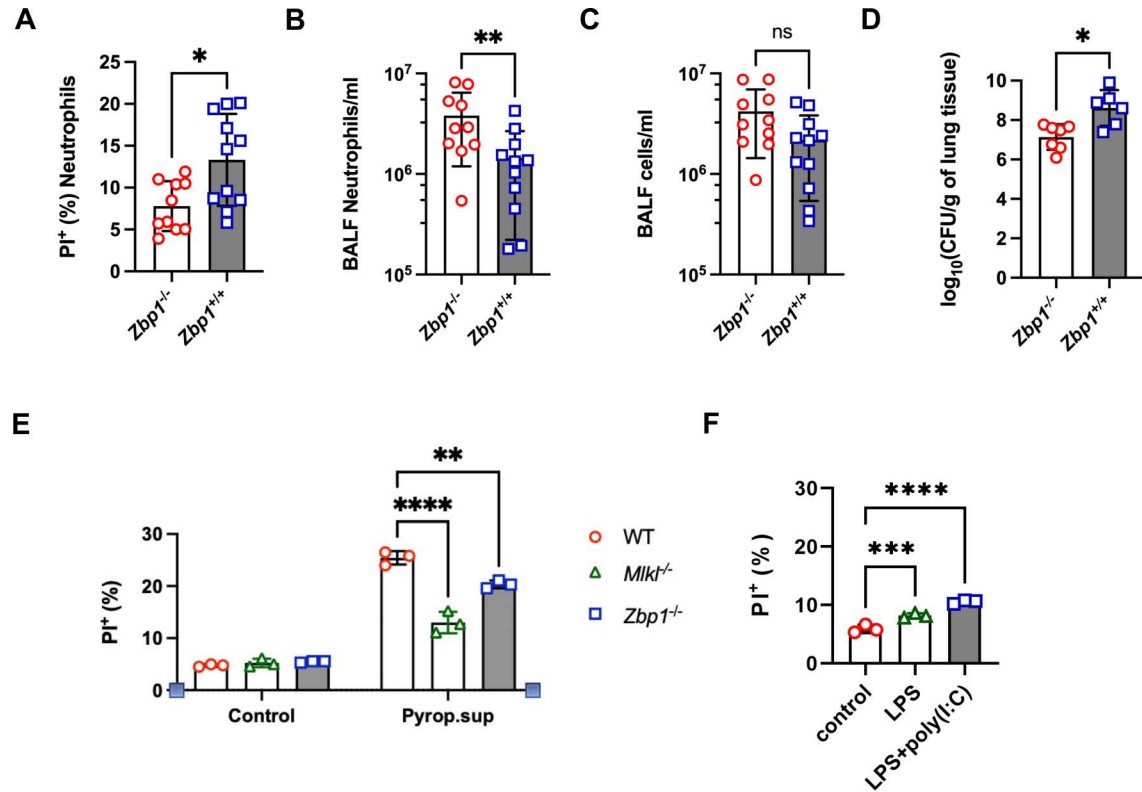

**Fig 6. ZBP1 deficiency protected against pulmonary *E. coli* infection.** (A-D) Twelve hours after *E. coli* infection, the neutrophil number (A), proportion of PI⁺ neutrophils (B), total leukocyte count (C) and bacterial burden (D) in the BALF of *Zbp1⁺/⁺* mice (n = 11) and *Zbp1⁻/⁻* mice (n = 10) were determined. (E) WT, *Mlkl⁻/⁻*, and *Zbp1⁻/⁻* neutrophils (n = 3) were treated with *E.* coli-stimulated epithelial or endothelial cell supernatant, and cell death was evaluated via flow cytometry. (F) WT neutrophils were treated with control medium, LPS (1 μg/ml) or LPS+poly(I:C) (1 μg/ml) for 4 hours, and neutrophil death was determined by PI staining and flow cytometry. The data are shown as the means ± SDs. Statistical differences were determined by Student's *t* test (A, B, C and D), two-way ANOVA (E) and one-way ANOVA (F). *$P < 0.05$; **$P < 0.01$; ***$P < 0.001$; ****$P < 0.0001$. ns, not significant. CFU, colony-forming unit. See also S9 Fig.

## Discussion

In the present study, we found that the activation of caspase-1 exacerbated the pathophysiology of pulmonary *E. coli* infection by driving infiltrating neutrophil necroptosis. Neutrophil necroptosis is not directly triggered in a cell-intrinsic manner by bacteria but is indirectly triggered by pyroptotic lung epithelium during *E. coli* infection. Moreover, pyroptotic epithelial cell-released dsRNA induced neutrophil necroptosis in a ZBP1-dependent manner. These findings revealed for the first time that two types of programmed cell death, controlled by distinct death-executing molecules, can communicate intercellularly during pathogenic infection.

Previous studies have reported that pyroptosis and necroptosis participate in the pathogenesis of bacterial pneumonia. Moreover, some studies have identified crosstalk between inflammasome activation and necroptosis that is mediated in a cell-intrinsic manner by a mechanism in which membrane-associated MLKL causes potassium efflux and induces NLRP3 activation [36]. However, the connection between these two types of programmed cell death in different types of cells remains to be determined. Our results showed that nonhematopoietic cell inflammasome-activated pyroptosis triggered necroptosis in infiltrating neutrophils. This result demonstrated the intercellular communication between pyroptosis and necroptosis, which contributes to the understanding of programmed cell death regulation and provides additional insights for future studies.

Cell death may be associated with many other immune events during pathogenic insult. LPS from gram-negative bacteria can induce cell death. Our results showed that LPS induced neutrophil p-MLKL but only very mild cell death. LPS combined with other substances in bacteria and host cells may collaborate to further promote neutrophil death. The release of damage-associated molecular patterns, including IL-1β, IL-18, HMGB1 and adenosine triphosphate, from cell rupture potentially amplifies inflammatory responses and mediates cell death [32,37,38]. Furthermore, programmed cell death-released substrates may support the growth of invasive microbes [39]. These paracrine effects of dead cells have important impacts on the development of various diseases. Moreover, we found that sterile epithelial cell content promoted the necroptotic membrane permeabilization in neutrophils with costimulation of bacterial secretion. These findings demonstrated that the epithelial cell content resulting from pyroptosis could induce necroptosis in neutrophils in a paracrine manner. Moreover, our results showed that the release of dsRNA by lung epithelial cells is one of the possible extracellular triggers of neutrophil necroptosis. We found that extracellular dsRNA and ZBP1-regulated neutrophil necroptosis could be potential targets for the treatment of these cell death-related diseases, including infectious diseases, cancer [40], neurological diseases [41], ischemic diseases and cardiovascular diseases [42].

This study has several limitations. First, a direct way to prove this cellular crosstalk *in vivo* is to utilize cell-specific gene manipulation in mice, including conditional knockout with the Cre-loxP system. We only used chimeric reconstructed mice in our *in vivo* bacterial pneumonia disease model. Second, bacterial load and inflammation are critical factors that can influence mouse survival during disease. Although caspase-1 modulates bacterial clearance and inflammation status, it remains unclear how bacterial load, inflammation, and neutrophil necroptosis individually contribute to mouse survival. Third, the initiating inflammasome upstream of caspase-1 in this model has not been investigated. Inflammasomes are crucial activators of caspase-1 and pyroptosis. Understanding this mechanism could provide valuable insights for manipulating the pathophysiology of bacterial pneumonia. Fourth, the mechanisms of dsRNA-triggered neutrophil ZBP1-regulated necroptosis are not fully detailed in this study due to technical limitations. This may involve one or a combination of the following processes: first, bacterial secretions form membrane pores to promote the uptake of dsRNA by neutrophils and ZBP1 sensing of extracellular substances, including dsRNA [43]; second, bacterial secretion stimulation upregulates the expression of neutrophil dsRNA receptors, such as SID-1 transmembrane family member 1 and SID1 transmembrane family member 2 [44,45], leading to greater dsRNA uptake and ZBP1 sensing. Third, according to our results, dsRNA uptake and sensing may not be the sole mechanism underlying neutrophil necroptosis, as dsRNA neutralization does not fully inhibit neutrophil death *in vitro*. There could be other regulatory mechanisms, including those involving Toll-like receptors or ATP sensing, that may also participate. Future studies are warranted to fully elucidate the crosstalk between programmed cell death pathways.

In summary, we showed that caspase-1 inflammasome-licensed pyroptosis could drive necroptosis in a paracrine manner during pulmonary *E. coli* infection. This communication is mediated by pyroptotic epithelial cell-released intracellular dsRNA, which is recognized by neutrophils and serves as a trigger for ZBP1-dependent necroptosis. These findings not only demonstrate a paradigm of communication between necroptosis and pyroptosis in different cell types during microbe invasion but also provide directions for manipulating immune defenses against infectious diseases.

## Materials/Subjects and methods

### Ethics statement

The animal experimental protocols were approved by the Laboratory Animal Welfare and Ethics Committee of Zhejiang University School of Medicine and were in compliance with institutional guidelines. This study did not involve human participants, human data or human tissue.

### Mice

C57BL/6 WT mice aged 6–8 weeks were purchased from Shanghai SLAC Laboratory Animal Corporation (Shanghai, China). *Casp1-/-* mice were kindly provided by Professor Di Wang of Zhejiang University School of Medicine. *Mlkl-/-* mice

and *Zbp1*[-/-] mice were obtained from Cyagen Biosciences (Suzhou, China), Inc. *Tlr3*[-/-] mice were gifts from Professor Chun-Feng Wang of Jilin Agricultural University [46]. Experiments involving gene-edited mice were performed with littermate controls, whereas experiments involving antibody or drug treatments were conducted using purchased, cohoused control mice. All the mice were housed in a specific pathogen-free and temperature-controlled standard environment in accordance with the National Institutes of Health Guide for Care and Use of Laboratory Animals. The animal experimental protocols were approved by the Laboratory Animal Welfare and Ethics Committee of Zhejiang University School of Medicine and were in compliance with institutional guidelines.

### Bacterial strains

*E. coli* bacteria (ATCC 25922) were obtained from the American Type Culture Collection (ATCC). The clinical strain of *E. coli* was isolated from a hospitalized patient with infectious arthritis of the knee at the Children's Hospital, Zhejiang University School of Medicine, Hangzhou, China. The strain was only used in a murine infection model to assess pathophysiology.

### Reagents and antibodies

For the inhibition experiments, the NEC-1 (HY-15760, 4311-88-0), GSK-872 (HY-101872, 1346546-69-7), z-VAD-FMK (HY-16658B, 161401-82-7), Ac-YVAD-cmk (HY-16990, 178603-78-6), GW806742X (HY-112292, 579515-63-2), and disulfiram (HY-B0240, 97-77-8) inhibitors were purchased from MedChemExpress. For immunoblot analysis, the following primary antibodies were used: caspase-1 (Abcam, ab179515), GSDMD (Abcam, ab209845), caspase-3 (Cell Signaling Technology, 14220), MLKL (Abcam, ab243142), MLKL (phospho S345; Abcam, ab196436), RIPK3 (Novus Biologicals, NBP1–77299), RIPK3 (phospho T231 + S232; Abcam, ab205421), HMGB-1 (Abcam, ab79823), dsRNA (SCICONS, 10010200) and β-actin (Sigma–Aldrich, A5316). The following substances were used as indicated: LPS (Sigma–Aldrich, L2630), TNF-α (BioLegend, 575202), Il-1β (PeproTech, 211-11b), and HMGB-1 (R&D Systems, 1690-HMB-050).

### Pulmonary *E. coli* infection model

*E. coli* bacteria were seeded on Luria–Bertani agar plates and cultured overnight, and a single clone was selected for culture in liquid Luria–Bertani medium and incubated at 200 rpm at 37 °C for another 12 hours. The bacterial suspension was then prepared at a concentration of $3 \times 10^6$ colony formation units (CFUs) per 50 μl of PBS. For pulmonary bacterial infection, 50 μl of live *E. coli* suspension was intratracheally administered to the mice. Mortality was assessed hourly. In some experiments, the mice were intraperitoneally injected with GW806742X (100 μM, 100 μl in PBS) or anakinra (100 mg/kg, 100 μl in saline; Bioxell, BE0246) or intratracheally injected with anakinra (200 μg/kg, 20 μl in PBS), an anti-dsRNA antibody (500 μg/kg, 20 μl in PBS), or related control solvent or control IgG (Bioxell, BE0260; Absin, abs20038) 1 hour before the induction of pulmonary *E. coli* infection.

### Cell count, protein level and bacterial load in BALF

Mouse lungs were washed with 0.5 ml of cold PBS three times, and the total white blood cell count was immediately calculated using a hemocytometer. The protein concentration was determined using a commercially available BCA protein assay kit (Thermo Fisher Scientific, 23225) according to the manufacturer's instructions. The bacterial load in the BALF was determined by adding the diluted BALF to Luria–Bertani agar plates and incubating at 37 °C overnight. The bacterial load in CFU per unit was calculated by multiplying the countable bacterial colony units by the number of dilutions.

### Collection of BALF cells and lung samples for western blot analysis

The mice were sacrificed 12 hours after *E. coli* infection. BALF was collected by performing three washes with 0.5 ml of PBS, followed by centrifugation at 800 × g for 5 minutes to isolate BALF cells. The lungs were subsequently harvested and

ground in liquid nitrogen. For protein extraction, lysis buffer (10 mM Tris-HCl, pH 7.4, 1 mM EDTA, 150 mM NaCl, 0.5% NP-40, 1 mM $Na_3VO_4$, and 1 mM PMSF) was added to both the BALF cells and the lung tissue. The samples were then subjected to ultrasonic fragmentation at 4 °C and 50 W for 3 minutes. Next, the samples were centrifuged at 12,000 rpm for 10 minutes, and the supernatants were collected for protein concentration analysis using a BCA protein assay kit. The samples were then subjected to SDS–PAGE and transferred to a polyvinylidene fluoride membrane (Millipore). The western blots were probed with the indicated antibodies and visualized via an EZ-ECL kit (Biological Industries, 20-500-120). Images were taken via a Clinx ChemiScope3300.

## Histology

Mouse lungs were harvested and fixed in 4% paraformaldehyde for at least 24 hours, embedded in paraffin wax and sectioned at a thickness of 5 μm. The sections were stained with hematoxylin and eosin (H&E) to analyze cellular and tissue morphological changes. Images were taken using an Olympus VS120 microscope (Shinjuku). Assessment of histological lung injury was performed by grading as previous described and the details are as follows: 1, normal; 2, focal (<50% lung section) interstitial congestion and inflammatory cell infiltration; 3, diffuse (>50% lung section) interstitial congestion and inflammatory cell infiltration; 4, focal (<50% lung section) consolidation and inflammatory cell infiltration; 5, diffuse (>50% lung section) consolidation and inflammatory cell infiltration [47,48]. For each section, at least five random high-power fields were selected for scoring. The mean score was used for comparison between groups. For TUNEL staining, mouse lungs were fixed in 4% paraformaldehyde for 12 hours at 4 °C, incubated in sucrose solution until the sample dropped to the bottom of the tube, embedded in optimal cutting temperature compound (Sakura, 4583) and sectioned at a 10-μm thickness. The sections were stained using a TUNEL kit (Beyotime, C1086) according to the manufacturer's instructions. For immunofluorescence staining of p-MLKL, BALF cells were air-dried on L-lysine-coated slides and stained with PKH26 (Sigma–Aldrich, MIDI26–1KT) according to the manufacturer's instructions. Then, the cells were fixed with 4% formaldehyde at room temperature for 5 minutes, permeabilized with 1% Triton X-100 for another 5 minutes, blocked with 10% bovine serum albumin (Sigma–Aldrich, A2153) at room temperature for 1 hour, and incubated with a primary antibody against p-MLKL at 4 °C overnight. The cells were then incubated with an AF647-conjugated secondary antibody (Thermo Fisher Scientific, A-31573) at room temperature for 1 hour. Nuclei were stained with 4',6-diamidino-2-phenylindole (Beyotime Biotechnology, P0131). Images were taken with an Olympus FV3000 microscope (Olympus), and the data were analyzed with ImageJ software (NIH).

## Flow cytometry for cell classification and cell death

For determination of the proportions of neutrophils and macrophages in the BALF, the cells were separated from the BALF by centrifugation at 800 × g for 5 minutes and resuspended in 100 μl of PBS. The cells were labeled with FITC-conjugated anti-mouse Ly6G/Ly6C (BioLegend, 108405), PE-CY7-conjugated anti-mouse F4/80 (eBioscience, 25-4801-82), APC-conjugated anti-mouse SiglecF (BioLegend, 155508), and BV421-conjugated anti-mouse Ly6C (BioLegend, 128031) for 25 minutes at 4 °C. The cells were subsequently washed and resuspended in flow cytometry buffer and stained with 2.5 μg/ml PI (Beyotime, ST511) for analysis.

For the caspase-1 activation assay in the lung parenchyma, the lungs of the mice were prepared as previously described. Briefly, mouse lungs were washed with 1 ml of HBSS with Liberase (100 μg/ml final concentration; Roche, 5401119001) and digested in 4 ml of HBSS digestion medium for 40 minutes at 37 °C with vortexing every 10 minutes. The resulting samples were filtered through a 70-μm cell strainer, washed with Dulbecco's modified Eagle's medium and treated with red blood cell lysis buffer (eBioscience, 00-4333-57) to lyse red blood cells. The cell suspensions were then stained with FITC-conjugated anti-mouse Ly6G/Ly6C, BV421-conjugated anti-mouse F4/80 (BioLegend, 123132), BV510-conjugated anti-mouse CD326 (BioLegend, 118231), APC-conjugated anti-mouse CD31 (BioLegend, 102509), FLICA 660-YVAD-FMK caspase-1 detection probes (ImmunoChemistry Technologies, 9122), and fixable viability dye 780

(BD Biosciences, 565388). Control isotype antibodies were used to determine the cutoff between negative and positive fluorescent populations. The data were collected with a BD LSRFortessa flow cytometer (BD Biosciences) and analyzed using FlowJo v.10 software (BD Biosciences).

## Macrophage or neutrophil depletion *in vivo*

For depletion of macrophages *in vivo*, the mice were intratracheally injected with 100 µl of clodronate liposomes or control liposomes (FormuMax Scientific, F70101C-NC-10) 48 hours before induction of the pneumonia model. For depletion of neutrophils *in vivo*, mice were intraperitoneally injected with 200 µg of anti-ly6G antibody (BioXCell, BE0075–1) or isotype control (BioXCell, BE0089) 2 days before the experiment and were intratracheally injected with 100 µg of anti-ly6G antibody or isotype control on the day of the experiment. The depletion efficacy was confirmed by flow cytometry.

## Bone marrow chimeric mice

WT, *Casp1*[-/-] and *Gsdmd*[-/-] mice were given gentamycin (0.08 g/200 ml) in the drinking water 1 week before radiation, which was stopped at 1 week after radiation. All the mice received 7 Gy of radiation for 20 minutes in an RS2000 Pro X-ray Biological Irradiator (Rad Source, US). The mice were retro-orbitally injected with bone marrow cells ($5 \times 10^6$) within 24 hours after radiation. Six to eight weeks later, the chimeric mice were used for further experiments.

## Cell culture

MLE12 cells (American Type Culture Collection, CRL-2110) were cultured in DMEM/F12 1:1 medium (Cytiva, SH30023.01) supplemented with 2% fetal bovine serum (FBS; Biological Industries, 04-001-1ACS) in an atmosphere with 5% $CO_2$ at 37 °C. The cells were seeded at $4 \times 10^5$/ml for further experiments. For siRNA silencing, MLE-12 cells were seeded at $1 \times 10^5$ in a 24-well plate and cultured. On the second day, Casp1 siRNA, GSDMD siRNA or control siRNA (10 pmol of each) was preincubated with Lipofectamine RNAiMAX Reagent (3 µl) for 5 minutes in Opti-MEM at room temperature as instructed. The mixture was then added to the cells, and the cells were incubated at 37°C for 72 hours for further analysis.

## Isolation of bone marrow neutrophils

For bone marrow neutrophils, the mice were sacrificed by cervical dislocation, and the femurs and tibias were harvested. The bone marrow cells were flushed from the femurs and tibias with DMEM supplemented with 10% FBS using a syringe with a 25-gauge needle. The cell suspension was collected and centrifuged at $800 \times g$ for 5 minutes at room temperature. Discontinuous Percoll (65% and 78%; Absin, abs9102) gradients were prepared and added to a 15-ml centrifuge tube according to the manufacturer's instructions. Pelleted cells were then resuspended in 3 ml of DMEM, gently added to the top of 65% Percoll solution and subjected to centrifugation at $800 \times g$ for 30 minutes. The cell layer between 65% and 78% was bone marrow neutrophils, and the purity was > 95% according to flow cytometry verification. The neutrophils were immediately used for subsequent experiments without further culture.

## Preparation of infectious cell supernatant, cell lysate and bacterial supernatant

For preparation of the infectious cell supernatant, live *E. coli* were added to MLE12 cells at a multiplicity of infection (MOI) of 2. The cell-free supernatant was collected 4 hours later and centrifuged at $3000 \times g$ for 5 minutes to remove cell debris and bacteria. The supernatant was subsequently filtered through a 0.2 µm syringe filter. This infectious cell supernatant or complete medium control was frozen at -20 °C for later use. For preparation of cell lysates, MLE12 cells were digested and resuspended in complete medium in 2-ml cryogenic storage vials and subjected to freeze–thaw cycles. For each freeze–thaw cycle, the cells were frozen in liquid nitrogen and thawed in a 37 °C water bath. The supernatant was

subsequently collected and adjusted to the appropriate culture volume using complete medium, after which it was centrifuged and filtered as described above. For preparation of the bacterial supernatant, *E. coli* was cultured at a concentration of $8 \times 10^5$ CFU/ml in complete medium as MLE12 cells. Four hours later, the supernatant was collected and processed as described above.

### *In vitro* stimulation and cell death analysis of neutrophils

*E. coli* (MOI = 2), LPS (1 µg/ml), heated *E. coli* (MOI = 20) or the collected supernatants were added to the neutrophils 1 hour after inhibitor (GSK-872, GW806742X, NEC-1 or Z-VAD FMK) treatment. Four hours after stimulation, the cells were centrifuged, and the lysates were collected for western blot analysis. For some experiments, the nuclei were stained with 2.5 µg/ml PI, and the data were collected with a Beckman Coulter DxFLEX flow cytometer (Beckman Coulter Life Sciences) and analyzed with FlowJo v.10 software. For immunofluorescence analysis, purified neutrophils were treated with LPS (1 µg/ml) and poly(I:C) (1 µg/ml) for 4 hours, and the cells were collected, stained with PKH26 and distributed on slides. After drying, fixation, and permeabilization, the cells were stained with anti-ZBP1 and anti-AF647 secondary antibodies (MedChemExpress) for analysis. Images were taken with an Olympus FV3000 microscope (Olympus), and the data were analyzed with ImageJ software (NIH).

### Cell death induction assays

Inhibitors (Ac-YVAD-cmk 40 µM, disulfiram 50 µM) and the control (dimethyl sulfoxide) were added to MLE12 cells 1 hour before bacterial infection. Live *E. coli* was added to MLE12 cells at an MOI of 2. At the indicated time points, the supernatant was collected and processed as described. PBS was added, and the nuclei were stained with 2.5 µg/ml PI. Images were taken with an Olympus IX73 inverted microscope. The red masks used for quantification of the PI stain are shown in the representative images.

### Enzyme-linked immunosorbent assay (ELISA)

The levels of the cytokines Il-1β (MultiSciences, EK201B) and TNF-α (MultiSciences, EK282) and the chemokines CXCL-1 (MultiSciences, EK296) and CXCL-2 (MultiSciences, EK2142) were measured using commercially available kits according to the manufacturers' instructions.

### Transcriptomic RNA-seq

Total RNA was isolated from BALF neutrophils from *Casp1*[-/-] mice and *Casp1*[+/+] mice using TRIzol reagent (Life Technologies, 155896018) following the manufacturer's protocol. RNA-seq was performed by LC-Bio Technology Co., Ltd. (Hangzhou, China) according to the manufacturer's recommendations. Briefly, total RNA quantity and purity were analyzed using a Bioanalyzer 2100 and an RNA 6000 Nano LabChip kit (Agilent), with RNA integrity values >7.0. Poly(A) RNA was purified from total RNA (5 µg) via two rounds of purification with poly-T oligo-attached magnetic beads. Following purification, the mRNA was fragmented into small pieces using divalent cations under elevated temperature. The cleaved RNA fragments were subsequently reverse-transcribed to create the final cDNA library in accordance with the protocol for TruSeq RNA Sample Preparation v.2 (Illumina, RS-122–2001 and RS-122–2002); the average insert size for the paired-end libraries was 300 bp (±50 bp). Paired-end sequencing was carried out on an Illumina NovaSeq 6000 following the manufacturer's recommended protocol. Before assembly, low-quality reads were removed using Fastp software (https://github.com/OpenGene/fastp). HISAT2 (https://ccb.jhu.edu/software/hisat2) was subsequently used to map reads to the reference genome of *Mus musculus* GRCm39. After the final transcriptome was generated, StringTie (https://ccb.jhu.edu/software/stringtie) was used to estimate the expression levels of all the transcripts. The differentially expressed mRNAs with a fold change > 2 or < 0.5 and with a parametric F test comparing nested linear models (*P* value < 0.05) were selected via the R package edgeR (https://bioconductor.org/packages/release/bioc/html/edgeR.html).

## Statistical analysis

The data are shown as the means ± SDs. Differences were analyzed by Student's $t$ test or ANOVA. The Mantel–Cox test and log-rank comparison were performed for survival experiments. A $P$ value < 0.05 was considered statistically significant. Statistical analysis was carried out using GraphPad Prism 9.3 (GraphPad Software).

## Supporting information

**S1 Fig. Lung neutrophils are critical for defense against *E. coli* pneumonia in *Casp1*-/- mice.** (A and B) Respiratory rates and body temperatures of *Casp1*-/- mice (n = 11) and *Casp1*+/+ mice (n = 8) at 12 hours after *E. coli* infection or intratracheal PBS instillation (n = 5 each). (C) BALF protein concentrations in *Casp1*-/- mice (n = 11) and *Casp1*+/+ mice (n = 8) at 12 hours after *E. coli* infection or intratracheal PBS instillation (n = 5 each). (D) BALF bacterial burden in *Casp1*-/- mice (n = 11) and *Casp1*+/+ mice (n = 8) at 12 hours after *E. coli* infection or intratracheal PBS instillation (n = 5 each) was determined and is expressed as CFU/ml. (E and F) IL-1β and TNF-α levels in the BALF of *Casp1*-/- mice (n = 11) and *Casp1*+/+ mice (n = 8) at 12 hours after *E. coli* infection or intratracheal PBS instillation (n = 5 each) were measured by ELISAs. (G and H) Resident macrophages (SiglecF+Ly6C-F4/80+) and recruited monocyte (SiglecF-Ly6C+F4/80+) in the BALF of *Casp1*-/- mice (n = 11) and *Casp1*+/+ mice (n = 8) at 12 hours after *E. coli* infection or intratracheal PBS instillation (n = 5 each) were counted by flow cytometry. (I) Total cellular ROS (determined by CellROX) and mitochondrial ROS (determined by MitoSOX staining) in neutrophils and macrophages from the BALF of *Casp1*-/- mice and *Casp1*+/+ mice at 12 hours after pulmonary *E. coli* infection were analyzed by flow cytometry (n = 7). (J) Pathway enrichment of expressed genes in neutrophils from the BALF of *Casp1*-/- mice and *Casp1*+/+ neutrophils at 12 hours after pulmonary *E. coli* infection was analyzed using RNA-seq transcriptomes (n = 2). The data are shown as the means ± SDs (A-I). Statistical differences were determined by two-way ANOVA (A–H), Student's $t$ test (I) and the parametric $F$ test comparing nested linear models (J). *$P$ < 0.05; **$P$ < 0.01; ***$P$ < 0.001; ****$P$ < 0.0001. ns, not significant. CFU, colony-forming unit.
(TIF)

**S2 Fig. Chemotaxis of neutrophils in the lung during pulmonary *E. coli* infection.** BALF cells (A) and BALF neutrophils (B) from *Casp1*-/- mice (n = 6) and *Casp1*+/+ mice (n = 6) were counted at 4 hours after bacterial instillation. Peripheral white blood cells (C) and neutrophils (D) from *Casp1*-/- mice and *Casp1*+/+ mice were counted at 12 hours after *E. coli* challenge (n = 11 each) or PBS instillation (n = 5 each). BALF levels of CXCL1 (E) and CXCL2 (F) in *Casp1*-/- mice and *Casp1*+/+ mice were measured at 12 hours after *E. coli* challenge (n = 11 each) or PBS instillation (n = 5 each). (G) LDH levels in the BALF of *Casp1*-/- mice and *Casp1*+/+ mice at 12 hours after *E. coli* infection (n = 6) or intratracheal PBS instillation (n = 5). The data are shown as the means ± SDs in (A-G). Statistical differences were determined by Student's $t$ test (A and B) and one-way ANOVA (C, D, E, F and G). ***$P$ < 0.001; ****$P$ < 0.0001. ns, not significant.
(TIF)

**S3 Fig. TUNEL staining of lung sections during pulmonary *E. coli* infection.** (A) TUNEL staining of lung sections from *Casp1*-/- mice and *Casp1*+/+ mice at 12 hours after *E. coli* infection and quantification of the TUNEL-positive area relative to the DAPI-stained area (n = 4). Scale bars, 100 μm. (B) *Casp1*+/+ and *Casp1*-/- mouse lungs were harvested at 12 hours after *E. coli* infection. TUNEL staining of lung sections was performed with different markers including the epithelial marker CD326, the endothelial marker CD31, the type II epithelial marker SPC, the macrophage marker F4/80, the neutrophil marker Ly-6G, and the monocyte marker CD14. Quantification of the percentage of TUNEL+ cells of different cell type (*n* = 3 biologically independent samples). Scale bars, 50 μm. The data are shown as the means ± SDs and are representative of 3 independent experiments. Statistical differences were determined by Student's $t$ test. *$P$ < 0.05. HPF, high-power field.
(TIF)

**S4 Fig. p-MLKL staining of lung sections and BALF cells during pulmonary E. coli infection.** (A) *Casp1*$^{+/+}$ and *Casp1*$^{-/-}$ mouse lungs were harvested at 12 hours after *E. coli* infection. p-MLKL staining of lung sections with the neutrophil marker Ly6G was performed by immunofluorescence. Scale bars, 50 μm. (B) BALF cells from *Casp1*$^{-/-}$ mice and *Casp1*$^{+/+}$ mice were fixed on slides, stained with PKH26 and p-MLKL antibodies and visualized by using confocal microscopy. Scale bars, 10 μm. The data are representative of 3 independent experiments. (TIF)

**S5 Fig. Intervention with MLKL decreased neutrophil death and protected mice against E. coli pneumonia.** (A) Representative images of lungs with H&E staining and quantification of lung injury scores in GW806742X (n = 5)- or control solvent (n = 6)-treated WT mice at 12 hours after pulmonary *E. coli* infection or intratracheal PBS instillation (n = 5 each). Scale bars, 100 μm. BALF protein concentration (B) and bacterial burden (C) in WT mice treated with GW806742X (n = 6) or control solvent (n = 6) at 12 hours after pulmonary *E. coli* infection or intratracheal PBS instillation (n = 5 each). (D) Representative images of lungs with HE staining and quantification of lung injury scores in *Mlkl*$^{-/-}$ (n = 9) or *Mlkl*$^{+/+}$ (n = 6) mice at 12 hours after pulmonary *E. coli* infection. Scale bars, 100 μm. (E) Twelve hours after *E. coli* infection, mouse lungs were harvested from *Mlkl*$^{-/-}$ or *Mlkl*$^{+/+}$ mice, and the bacterial burdens in whole-lung tissue were determined. The data are shown as the means ± SDs (A, B, C, D and E). Statistical differences were determined by Student's *t* test (D and E) and two-way ANOVA (A, B and C). *$P < 0.05$; **$P < 0.01$; ****$P < 0.0001$. CFU, colony-forming unit. (TIF)

**S6 Fig. Neutrophil necroptosis is independent of bacterial stimuli and inflammatory stimuli.** (A) Proportion of PI$^+$ neutrophils from *Casp1*$^{-/-}$ and *Casp1*$^{+/+}$ mice after stimulation with LPS, live *E. coli,* heat-inactivated *E. coli* or PBS (n = 3). (B) Immunoblot analysis of cell death-related proteins in neutrophils stimulated with LPS, live *E. coli* or heat-inactivated *E. coli*. (C) Proportion of PI$^+$ neutrophils from WT mice treated with IL-1β, TNF-α, HMGB-1 or control solvent at the indicated concentrations (n = 3). (D and E) Survival of WT mice intraperitoneally (n = 5 each) or intratracheally (n = 6 each) pretreated with the IL-1β antagonist anakinra or saline after pulmonary *E. coli* infection. (F) Survival of WT mice intraperitoneally (n = 6 each) pretreated with the anti-IL-1β antibody or control IgG antibody after pulmonary *E. coli* infection. The data are shown as the means ± SDs in (A and C) and are representative of 3 independent experiments in (B). Statistical differences were determined by the Mantel–Cox test (D, E and F). ns, not significant. (TIF)

**S7 Fig. Pyroptotic epithelial cells trigger neutrophil necroptosis.** (A) Flow cytometry gating strategy for identifying caspase-1-activated cells in mouse lungs via FLICA staining. (B) MLE-12 epithelial cells were stimulated with *E. coli*, and then, the cell lysate and culture supernatant were collected and immunoblotted for pyroptosis-related proteins. (C and D) *Casp1*$^{-/-}$ neutrophils and *Mlkl*$^{-/-}$ neutrophils were pretreated with GSK872, GW806742X or control solvent and then stimulated with pyroptotic supernatant (pyrop. sup) or control medium, and cell death was assayed by flow cytometry (n = 3). (E) WT neutrophils were pretreated with NEC-1, Z-VAD or control solvent and then stimulated with pyrop. sup or control medium, and cell death was assayed by flow cytometry (n = 3). The data are shown as the means ± SDs in (C, D and E) and are representative of 3 independent experiments in (B). Statistical differences were determined by two-way ANOVA (C and D) or one-way ANOVA (E). ****$P < 0.0001$. (TIF)

**S8 Fig. Pyroptosis-released dsRNAs induce neutrophil necroptosis via ZBP1 during pulmonary E. coli infection.** The total BALF cell number (A), protein concentration (B) and bacterial burden (C) were measured in the mice pretreated with the anti-dsRNA antibody (n = 9) or the IgG control antibody (n = 10) at 12 hours after pulmonary *E. coli* infection. (D) Representative images of lungs with HE staining and quantification of lung injury scores in anti-dsRNA antibody (n = 6)- or control IgG (n = 6)-treated WT mice at 12 hours after pulmonary *E. coli* infection. Scale bars,

100 μm. (E–G) Proportion of PI+ neutrophils in the BALF of *Casp1⁻/⁻* (L), *Mlkl⁻/⁻* (M) and *Zbp1⁻/⁻* (N) mice treated with an anti-dsRNA antibody (n = 6–8) or an IgG control antibody (n = 7–8) at 12 h after pulmonary *E. coli* infection. (H) Representative images of lungs with HE staining and quantification of lung injury scores in *Zbp1⁻/⁻* (n = 6) or *Zbp1⁺/⁺* littermate (n = 6) mice at 12 hours after pulmonary *E. coli* infection. Scale bars, 100 μm. The data are shown as the means ± SDs. Statistical differences were determined by Student's *t* test. *$P < 0.05$; **$P < 0.01$. ns, not significant. CFU, colony-forming unit.
(TIF)

**S9 Fig. TLR3 may not be involved in the neutrophil necroptosis pathway during pulmonary *E. coli* infection.** (A) Western blot analysis of BALF neutrophils and spleen cells derived from *E. coli*-infected *Casp1⁻/⁻* and *Casp1⁺/⁺* mice at 12 hours. (B-E) Twelve hours after *E. coli* infection, the proportions of PI+ neutrophils (B), neutrophil numbers (C), total leukocytes (D) and bacterial burdens (E) in the BALF of *Tlr3⁺/⁺* mice (n = 8) and *Tlr3⁻/⁻* mice (n = 7) were determined. The data are shown as the means ± SDs (B-E) and are representative of 2 independent experiments in (A). Statistical differences were determined by Student's *t* test (B-D). ns, not significant. CFU, colony-forming unit.
(TIF)

**S1 Data. Quantitative data used in calculations corresponding to primary figures.** Numeric values used to generate graphs, means, and standard deviations for primary figures are included on tabs, with each tab indicating the relevant figure panel.
(XLSX)

**S2 Data. Quantitative data used in calculations corresponding to supplemental figures.** Numeric values used to generate graphs, means, and standard deviations for supplemental figures are included on tabs, with each tab indicating the relevant figure panel.
(XLSX)

## Acknowledgments

We thank Yanwei Li, Zhaoxiaonan Lin, Chun Guo and Jiajia Wang from the Core Facilities, Zhejiang University School of Medicine, for their technical support. We also thank Dr. Yang Yang from Sir Run Run Shaw Hospital, Zhejiang University School of Medicine, for help for *Tlr3⁻/⁻* mice.

## Author contributions

**Conceptualization:** Qinyu Luo, Qixing Chen.

**Formal analysis:** Qinyu Luo.

**Funding acquisition:** Qixing Chen.

**Investigation:** Qinyu Luo, Lihua Shen, Yan Zhang, Yihang Pan, Zehua Wu.

**Methodology:** Qinyu Luo, Lihua Shen, Shiyue Yang, Yan Zhang, Yihang Pan, Zehua Wu.

**Project administration:** Qiang Shu, Qixing Chen.

**Supervision:** Qiang Shu, Qixing Chen.

**Validation:** Zehua Wu.

**Visualization:** Qinyu Luo.

**Writing – original draft:** Qinyu Luo.

**Writing – review & editing:** Qixing Chen.

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
