## [Decision Letter · Decision Letter 0]

20 May 2024

Dear Professor Chen,

Thank you very much for submitting your manuscript "Caspase-1-licensed pyroptosis drives dsRNA mediated necroptosis and dampen host defense against bacterial pneumonia" for consideration at PLOS Pathogens. As with all papers reviewed by the journal, your manuscript was reviewed by members of the editorial board and by several independent reviewers. In light of the reviews (below this email), we would like to invite the resubmission of a significantly-revised version that takes into account the reviewers' comments.

We cannot make any decision about publication until we have seen the revised manuscript and your response to the reviewers' comments. Your revised manuscript is also likely to be sent to reviewers for further evaluation.

Sincerely,

Andrew J Monteith, Ph.D.

Guest Editor

PLOS Pathogens

Matthew Wolfgang

Section Editor

PLOS Pathogens

Michael Malim

Editor-in-Chief

PLOS Pathogens

orcid.org/0000-0002-7699-2064

Reviewer's Responses to Questions

**Part I - Summary**

Reviewer #1: Luo et al. use a E. coli respiratory infection mouse model of bacterial pneumonia to evaluate the host determinants of pathogenesis. Interestingly, the authors observe that Casp1 knockout (KO) mice experience less disease despite an increase in BALF neutrophils. These observations were largely phenocopied by Zbp1 and Mlkl KO mice deficient in necroptosis, raising the interesting question of how pyroptotic and necroptotic pathways intersect to promote lung disease. A combination of in vivo and in vitro experiments suggests a possible role for host-derived dsRNA in the initiation of Zbp1-dependent necroptosis in neutrophils. By evaluating FLICA and cell death in distinct cell types from infected lungs, the authors observed enhanced Casp1 activity primarily in epithelia (and endothelia) whereas markers of necroptotic death were primarily observed in neutrophils. Based on these findings and some clever in vitro experiments, the authors conclude that E. coli lung infection causes lung epithelial cell Casp1-dependent pyroptosis, which releases dsRNA that is taken up by neutrophils leading to Zbp1-dependent necroptosis, the subsequent loss of neutrophils, and enhanced pathogenesis. In many respects, I find this model of non-intrinsic iterative cell death appealing, and it provides an important alternative mechanism to so called 'PANoptotic' cell-intrinsic pathways to explain how distinct cell death pathways can interact. However, many of the most exciting and new conclusions are largely based on correlative analyses that leave several alternative explanations for the data presented.

Reviewer #2: In “Caspase-1-licensed pyroptosis drives dsRNA mediated necroptosis and dampen host

defense against bacterial pneumonia”, the authors seek to show that caspase-1 mediated death of epithelial cells leads to necroptosis of neutrophils that negatively impacts lung health. Points of scholarship that would benefit from improvement: 1) Current epidemiological and other background (some studies are 20 years out of date and not accurate), 2) there is an insufficient description of E coli pneumonia as a disease, 3) the endotoxin responses in the lung, which has extensive literature that should be cited and discussed, since this has not been excluded as a major driver of inflammation in this model, and is expected to be. The topic is of overall interest with moderate novelty, using mostly appropriate methodology, but lacks support for major claims or relies on indirect evidence, building an overall model that could be incorrect. These are outlined below:

Reviewer #3: The manuscript by Qinyu Luo et al describes how epithelial cells-neutrophils crosstalk leading to the progression of pneumonia. Specifically, it showed during E. coli lung infection (pneumonia), Caspase1-mediated pyroptotic epithelial cells release dsRNA in the bronchoalveolar lavage fluid (BALF). As a result, dsRNA along with E. coli induce neutrophil necroptosis, which is associated with a detrimental impact on the disease outcome. The manuscript is well written except for the last portion of the result section. It used several genetic and pharmaceutical tools to support these findings. For the most part, I found the data supported the authors’ concussions. I think the findings are valuable to the field of lung immunopathology. The data will increase our understanding of how different death pathways are engaged in a complex environment with multiple types of cells to drive disease pathology. However, I have several concerns that could clear some of the ambiguous conclusions that the authors draw. The following are my concerns:

**Part II – Major Issues: Key Experiments Required for Acceptance**

Reviewer #1: One of the major claims is that Casp1-dependent pyroptosis is primarily occurring in epithelial cells, whereas Zbp1/Ripk3/Mlkl-dependent pyroptosis is primarily occurring in neutrophils downstream of epithelial pyroptosis. As the authors correctly point out, the in vivo experiments do not allow for this strong conclusion.

o Cell-type specific KO or KI mice (e.g., deletion of Casp1 and Mlkl in epithelial vs neutrophils) to assess these claims would be welcome and definitive, but I recognize the considerable time and cost associated with such experiments. Nevertheless, the authors focus primarily on epithelia, which ignores the result in Fig. 4A and 4B that Casp1 is active in endothelia. Thus, it is unclear whether the in vivo source of pyroptosis (and the presumed release of dsRNA) is epithelial, endothelia, or both.

o Reciprocal bone marrow chimera experiments would be an appropriate alternative. If the authors choose to do so, they might consider also including Gsdmd KO mice since they formally did not genetically demonstrate the role of pyroptosis in vivo or in vitro.

o While I find the in vitro experiments with MLE12 and neutrophils a useful in vitro model, the authors could much more rigorously support their claims by genetically deleting genes relevant to the inflammasome and necroptotic pathways in both MLE12 and neutrophils. Editing primary neutrophils may not be possible, but presumably neutrophils could be used from KO mice in this study, or alternatively, make use of more genetically tractable cell lines (e.g., Cas9+ER-Hoxb8 neutrophil progenitor). Such a model would also allow for complementation of KO neutrophils with variants of interest (e.g., Zbp1 mutants that lack the capacity to bind dsRNA).

The role for dsRNA in neutrophil necroptosis relies primarily on two observations: 1) that Zbp1 KO mice are protected from disease, and 2) injection of a dsRNA binding antibody attenuates disease. This is a remarkable claim, and is the key mechanistic detail linking the two cell death modalities in a non-cell intrinsic manner. Thus, further validation either in vivo or in vitro is important to substantiate the author’s conclusions:

o A prediction of the model is that the injection of dsRNA binding Ab would have no effect on a Zbp1, Ripk3, or Mlkl KO mouse, or similarly, in the response of neutrophils from those mice ex vivo.

o Epithelial cell death is not sufficient to induce neutrophil death, but an explanation is not provided. The authors should present data to determine if dsRNA is released from non-E. coli infected epithelia, and if so, if dsRNA is being taken up by neutrophils in the absence of bacteria. Similarly, the experiments in Figure S4A (e.g., WT vs Casp1 KO challenged with E. coli, heat-killed E. coli, etc) should be conducted on MLE12 cells, followed by conditioned supernatants from those studies being used to evaluate necroptosis in neutrophils derived from WT and relevant KO animals. Moreover, the authors rule out a role for IL-1B, but it remains possible that IL-18 or other signals released from pyroptotic epithelia are required to stimulate neutrophils for either the uptake of dsRNA or activation of the necroptotic pathway.

Reviewer #2: While E. coli pneumonia can occur, the strain used, ATCC 25922, is a non-pathogen. A relevant virulent clone known to cause pulmonary disease in humans or experimental models should be used.

Furthermore, there are no counts of bacteria in the lung for nearly experiment. This is critical to include throughout, and is very unconventional to exclude. A major limitation to the interpretation that the authors should additionally discuss how the greater survival of casp1-/- mice, and their lessor injury, could be due to them just having fewer bacteria (Fig S1). In this sense, inflammatory responses are typically proportional to bacterial load, and the effects reported could be entirely a consequence of that. Effects on IL-1b, respiration, temperature, neutrophil number, and their necroptosis, would all expect to be consequent to that, and not necessarily through the described casp-1-dependent mechanism.

The authors report that IL-1 is not involved because anakinra could not rescue the mice from lethal pneumonia (line 275). However, the authors have not shown that anakinra as delivered was available in the lung, nor that it inhibited IL-1 signaling like claimed. Actual signaling of IL-1 in the lung should to be measured to know whether the drug is having the stated effect, and this mechanism needs to be confirmed using IL-1R KO mice.

Similarly, the authors need to show that the cell ablations of Fig 1F and 1G are leading to actual depletion of these cells in the experiment.

The experiments of Fig 5F and 5G are an unconvincing demonstration of PolyIC uptake. F shows association with cells, which could be sticking on the surface and not uptake. G is very low magnification, and appears to show staining of the whole cell for some cells, and no staining at all for others, which isn’t a demonstration of colocalization. Higher-magnification imaging, with membrane markers if needed, should be used to show polyIC is intracellular, and colocalization requires that there are areas that there are areas lacking both polyIC and ZBP1, not just the whole cell imaged as positive for both.

Figure 2E compares proteins in BALF cells, which are mostly neutrophils, to the tissue, which is mostly not. For one, the methods for this experiment are unclear, not in the text, the legend, or figure is it stated how and when these samples were derived. I will assume this is just an omission and that they were infected mice, though not stated, and for a similar amount of time as other experiments. However, there are several biases in these comparisons needing controls including uninfected mice and purified neutrophils, and a survivorship bias, as their other data show evidence of significant death by this point, so it is living cells that have p-MLKL, arguing that the dead cells might have died by other means. This major point also needs the good blotting control using the MLKL-/- mice of Fig 3G

3E requires PBS control for infection-independent effects of the drug on neutrophil death

The cytometry of Fig 4A 4B requires additional details on markers, compensation, and gating, total and live/dead cells of each population, and controls of uninfected mice. As it is, it is not clear how abundant the cell populations are, how accurate their identification is, and how reliable the FLICA staining is. In particular, the cell processing method is quite harsh and expected to lead to significant cell death, which can bias these populations, and give false positives/negatives.

A statistician should review the methods, which are applied inconsistently between figures. For example, two-way ANOVA with Tukey-Kramer’s post hoc analysis is done for Fig 3 (C, D, E and F), but some of these are pairwise, others multivariate, and methods are applied inconsistently between different figures of similar data.

There are gaps in the claims on the specificity of each cell death & sensing mechanism to each cell type. This is most cleanly and directly addressed by lineage-specific knockout. It minimum, these limitations should be discussed and redundancies between these pathways should be excluded including the effect of dsRNA antibody on neutrophil death in infected casp1-/- and mlkl-/- mice, the effect of mlkl-/- on FLICA cells, and histological examinations added to the major mechanistic points (not just Fig 1B)

Reviewer #3: 1. The presented data did not directly demonstrate in vivo that pyroptotic epithelial cells are responsible for dsRNA release in BALF. It is also not clear which cell death (Pyroptosis versus necroptosis) occurs first, or whether dead neutrophil causes epithelial cell damage and release of dsRNA. To address these, the authors should use epithelial cell-specific Caspase-1 deficient mice. However, generating such a mouse line will take at least 6 months. Instead, adoptive transfer experiment could provide some support to the conclusions. Bone marrow-derived neutrophils from wild-type mice could be adoptively transferred to UV-irradiated Caspase-1 KO mice and test pneumonia outcomes (mice survival; bacterial and immune cell counts; dsRNA level in the BALF and lungs; neutrophil and epithelial cell death and so on). The mice should be protected if the proposed conclusion is correct (i.e., pyroptotic epithelial cells released dsRNA to induce necroptotic neutrophils). Also, bone marrow-derived neutrophils from Caspase-1 KO mice could be adoptively transferred to UV-irradiated wild-type mice and test pneumonia outcomes. The experiment outcome could assess whether epithelial cell death or neutrophil death occurs first.

2. Which type of epithelial cells are affected? the authors should combine Tunel staining of histology sections from wild-type and Caspase1 KO with different cell markers including type I epithelial, type II epithelial, macrophages, neutrophils, and monocytes.

3. A combination of LPS + poly I:C stimulation to induce necroptotic neutrophils in vitro and in vivo should be tested. This will provide evidence that dsRNA from dying cells along with LPS from bacteria are sufficient to derive necroptotic neutrophils.

4. The authors need to measure bacterial loads in the lungs. Measuring bacterial counts in the BALF is not sufficient to conclude bacterial clearance. Bacterial counts in BALF depend on the efficiency of performing the lavage (do the authors use equal volume to lavage mice and whether end up with equal volumes of BALF from wild type and KOs. Also, immune cell counts in the lungs are a better predictor of disease outcomes than BALF.

5. Fig. 1D showed a massive loss of macrophage during infection. Are they dying? If so, which cell deaths are responsible, and could they contribute to dsRNA release? Why there are lower macrophage counts in uninfected wild-type relative to Caspase-1 KO? Do Caspase-1 KO mice experience some sort of low-grade inflammation, especially in the lungs?

6. All immune cells count, and bacterial burden should be graphed using a log y-axis scale. This way we could see the lower number of immune cells and bacteria that are present in some of the conditions.

7. It is not clear whether BALF cells in Fig. 2C or Fig. 4H that induced necroptosis are macrophages, monocytes, or neutrophils. Although neutrophil counts are higher in BALF, the amount of protein in these cells is very low compared to macrophages and monocytes. The authors should either sort these cells or magnetic purify them. Similarly, the remnant lung tissue after lavage is not only epithelial cells. Many other cells including interstitial macrophages could be the driver for activation of Caspase-1 and GSDMD.

8. Treated mice with control solvents such as DMSO, Saline, control IgG…etc are dying at a faster rate than untreated mice. Examples include in fig. 3A, fig. 3B, fig.5I, supplementary fig. 4D and 4E. Do the authors know why this is the case? Does DMSO or saline treatment make the mice more susceptible to infection? This should be emphasized.

**Part III – Minor Issues: Editorial and Data Presentation Modifications**

Reviewer #1: - It is unclear why preventing neutrophil necroptosis limits pathogenesis. Do the authors consider this to be similar to pathology associated with NETosis?

- Can the authors provide an explanation based on their model for why E. coli CFU is lower in all KOs or interventions that prevent pyroptosis and necroptosis?

- I find it somewhat surprising that Escherichia coli Seattle 1946 strain induces epithelial cell Casp1-dependent death. Presumably this strain is attaching/effacing, and not replicating intracellularly? Can the authors comment on this and the nature of the upstream sensor that induces Casp1 activation in lung epithelia?

- Similarly, while I do not think it is within the scope of the present work, it would be interesting to know the inflammasome-forming sensor upstream of Casp1. This would increase the impact of the manuscript.

- Tracheal infection of E. coli is not very physiologic. Demonstrating this pathway occurs in the context of a respiratory pathogen would increase relevance.

- In Fig. S4B, Casp1 does seem to be active in neutrophils upon challenge with E. coli or LPS. Can the authors speak to this?

- For each mouse experiment, please clarify if mice were littermates, cohoused, or neither.

Reviewer #2: none

Reviewer #3: 1. Impact neutrophil depletion on wild-type mice should be provided. Does neutrophil depletion impact bacterial clearance in wild-type mice?

2. The lung injury score graph (Fig. 1B) should be fixed back to a standard y-axis graph. It is not clear why a two-segment y-axis was used.

3. The last section of the results was poorly written (lines 325-381). There are a lot of missing spaces, the authors used “we next” several times to start a new sentence. It seems that the authors listing experiment findings without rationale or context.

4. In line 41 “conbined” should be fixed to combined.

5. The authors should immunofluorescence stain other BALF and lung cells with pMLKL antibody, like supplement fig. 2H. This could provide evidence that necroptosis impacted only neutrophils and/or other cells.

6. Does pyroptosis still occur in epithelial cells when the mice were treated with GW806742X or in Mlkl KO mice?

7. It is not clear how purified neutrophils are dying in response to E. coli and heat-killed E. coli in supplementary fig. 4A. The authors need to include more detail on how infections were performed. What is the multiplicity of infection (MOI) used and what time point post-infection cell death was analyzed? Are extracellular bacteria washed or kept in the media? Also, LPS-treated purified neutrophils induce activation of MLKL but no cell death. The authors need to explain why LPS stimulation is insufficient to trigger necroptosis.

8. The number of mice used in the supplementary fig. 4D and 4E is low. It is not clear how many times the experiment was repeated on different days. Power analysis should be performed, and I recommend using a biostatistical expert to determine this.

9. For the in vivo data, the authors often list the n number, which represents the number of used mice. However, whether the experiments were done on different days needs to be clarified.

10. The authors should calculate and graph the intensity of bands from immunoblots in Fig. 2D and 2E.

11. What is the stain used in Fig. 4D? The authors should indicate detailed information in the figure legend and the method section.

12. It is not stated how the anti-dsRNA was administered to mice.

13. It is not clear what is the purpose of measuring long terminal repeat elements (LTRs), long interspersed nuclear elements (LINEs), or short interspersed nuclear elements. The authors need to explain the rationale behind doing such an analysis. Is the dsRNA measured by RNAseq originate from epithelial or neutrophil?

14. The authors state “However, GW806742X is not a specific inhibitor of MLKL” What other proteins does the inhibitor target?

PLOS authors have the option to publish the peer review history of their article (what does this mean? ). If published, this will include your full peer review and any attached files.

**Do you want your identity to be public for this peer review?** For information about this choice, including consent withdrawal, please see our Privacy Policy .

Reviewer #1: No

Reviewer #2: No

Reviewer #3: No
---

## [Decision Letter · Decision Letter 1]

23 Dec 2024

PPATHOGENS-D-24-00639R1

Caspase-1-licensed pyroptosis drives dsRNA mediated necroptosis and dampen host defense against bacterial pneumonia

PLOS Pathogens

Dear Dr. Chen,

Thank you for submitting your manuscript to PLOS Pathogens. After careful consideration, we feel that it has merit but does not fully meet PLOS Pathogens's publication criteria as it currently stands. Therefore, we invite you to submit a revised version of the manuscript that addresses the points raised during the review process.

Please submit your revised manuscript within 30 days Feb 21 2025 11:59PM. If you will need more time than this to complete your revisions, please reply to this message or contact the journal office at plospathogens@plos.org. Please include the following items when submitting your revised manuscript:

We look forward to receiving your revised manuscript.

Kind regards,

Andrew J Monteith, Ph.D.

Guest Editor

PLOS Pathogens

Matthew Wolfgang

Section Editor

PLOS Pathogens

Sumita Bhaduri-McIntosh

Editor-in-Chief

PLOS Pathogens

orcid.org/0000-0003-2946-9497

Michael Malim

Editor-in-Chief

PLOS Pathogens

orcid.org/0000-0002-7699-2064

**Additional Editor Comments :**

I appreciate the effort that the authors gave to resolving the issues associated with this manuscript; however, there are still concerns that exist or remain unaddressed.

• I share a similar concern that bacterial burdens in BALF may not be reflective of whole tissue burdens, which was only addressed by citing a reference in the rebuttal letter. However, my larger concern is the lack of burden quantification in the in vivo models, in particular when quantifying in vivo functions. Burden quantifications are critical to ensure that the infection was consistent and that identified phenotypes are not associated with differences in inoculations or infection dynamics. Every in vivo characterization should be coupled with CFU burdens to ensure accuracy of the data.

• Multiple pieces of data were provided in the rebuttal letter to address reviewer concerns, but these were not included in the manuscript (TUNEL staining with different cell populations, stimulation with LPS+ poly I:C, and pMLKL immunofluorescence stain of BALF cells). The point of these data is making the manuscript better not to directly address reviewers. This was an oversight not integrating these data into the manuscript. There were also instances in the text where conclusions are made without showing data (eg: line 161; phagocytosis efficiency…were comparable).

• The interpretation of Figure 5B is confusing. It seems unlikely that simply adding bacterial supernatant and MLE12 cell lysate nearly fully recreates the supernatant from pyroptotic MLE12 cells. These are entirely different stimulations. Also, if the stimulating agent is dsRNA that gets released from the epithelial cells, does pyroptosis cause an abundance of dsRNA? It is unclear why dsRNA (which would likely be in low abundance) would be such a potent stimulation when the abundance of cytokines and other nuclear molecules (dsDNA, ssRNA) would be much higher. In addition, why would dsRNA only be found in the BALF, but not identified in the lung homogenates (Fig. 5E)?

• The authors use an antibody to neutralize dsRNA during infection (Fig. 5I-K), but rather than simply neutralizing the dsRNA, this likely created immune complexes and would skew the inflammatory environment. It's really uncertain what the dsRNA antibody is doing, especially in vivo. A cleaner experiment would be to use TLR3-deficient mice, which are readily available, and prevent signaling by dsRNA. TLR3 seems the most likely receptor mediating neutrophil necroptosis and it is unusual that this was overlooked in the manuscript.

Data quality concerns:

• In Figure 5B the input lane is bands while the IP lanes look highly smeared or dot blot-like (ZBP1 bands). Presumably this was all run on the same gel, so why are the lanes so variable in presentation?

• The fluorescent images provided in Figure 5G are not of sufficient quality to draw any conclusions. The pattern of the dapi stain seems unusual and doesn’t seem to be staining the nucleus. The background of the Poly(I:C) in the bacterial stimulated cells looks excessive and localized to a specific region, which is not present in the mock stimulated. Despite these concerns in the images, providing a single cell is not sufficient to draw conclusions. Higher quality representative images and quantification is required.

• Supplementary Figure 2I includes images of BALF cells with high pMLKL protein at the membrane. Based on the proposed model, the neutrophils are the cell type with high pMLKL protein; however, the nuclear staining (Dapi) of these cells do not look neutrophil-like. This is concerning and requires further validation (Ly6g or neutrophil specific protein staining) to ensure the cell type of the BALF cells.

**Journal Requirements:**

1) We note that your Data Availability Statement is currently as follows: "All data generated and analyzed during this study are available." Please confirm at this time whether or not your submission contains all raw data required to replicate the results of your study. Authors must share the “minimal data set” for their submission. PLOS defines the minimal data set to consist of the data required to replicate all study findings reported in the article, as well as related metadata and methods (https://journals.plos.org/plosone/s/data-availability#loc-minimal-data-set-definition).

Please ensure that the Data Availability Statement provided in the online submission form matches the one mentioned in the manuscript.

2) Please amend your detailed Financial Disclosure statement. This is published with the article. It must therefore be completed in full sentences and contain the exact wording you wish to be published.

1) Please clarify all sources of financial support for your study. List the grants, grant numbers, and organizations that funded your study, including funding received from your institution. Please note that suppliers of material support, including research materials, should be recognized in the Acknowledgements section rather than in the Financial Disclosure.

2)State the initials, alongside each funding source, of each author to receive each grant. For example: "This work was supported by the National Institutes of Health (####### to AM; ###### to CJ) and the National Science Foundation (###### to AM)."

3) State what role the funders took in the study. If the funders had no role in your study, please state: "The funders had no role in study design, data collection and analysis, decision to publish, or preparation of the manuscript."

3) Your current Financial Disclosure states, "The author(s) received no specific funding for this work."

However, your funding information on the submission form indicates receiving a fund. Please ensure that the funders and grant numbers match between the Financial Disclosure field and the Funding Information tab in your submission form. Note that the funders must be provided in the same order in both places as well.. 

Please indicate by return email the full and correct funding information for your study and confirm the order in which funding contributions should appear. Please be sure to indicate whether the funders played any role in the study design, data collection and analysis, decision to publish, or preparation of the manuscript.

**Comments to the Authors:**

Please note that one of the reviews is uploaded as an attachment.

**Reviewers' Comments:**

Reviewer's Responses to Questions

**Part I - Summary**

Reviewer #1: The authors have addressed my major issues, in particular, the bone marrow chimeras indicating that pyroptosis and necroptosis occurs in distinct compartments (hematopoietic vs non-hematopoietic, respectively). In my opinion, this constitutes a significant advance in our understanding of how distinct cell death pathways in distinct cell types can underlie the host response and pathogenesis in a model of respiratory bacterial infection.

Reviewer #3: The authors addressed some of the first round of reviews. However, there are still many concerns that need to be addressed. I also recommend having English as the first language colleague and working in the same field to edit the manuscript.

**Part II – Major Issues: Key Experiments Required for Acceptance**

Reviewer #1: (No Response)

Reviewer #3: 1) The authors did not include the new data (TUNEL staining, stimulation with LPS+ poly I:C, and pMLKL immunofluorescence stain of BALF cells) in the revised manuscript. Only zoomed-out immunofluorescence images were provided, which made it difficult to analyze. The authors should provide higher-resolution zoomed-in images and quantify the data.

2) Measuring bacterial load in the lungs is still missing. Providing references that used BALF bacterial burden to assess lung pneumonia is weak. The authors should attempt at least to provide CFU data in the lungs of WT and Casp1 KO mice.

**Part III – Minor Issues: Editorial and Data Presentation Modifications**

Reviewer #1: The authors should quantify results shown in figures Fig. 5F and 5G.

Reviewer #3: 1) In the Abstract (line 18), what did the authors mean by “cell reaction”? Need some clarification.

2) In line 38, there is a missing period at the end of the sentence.

3) How is regulated cell death participating in the pathological process as stated in line 39? What do the authors mean by stating “regulated cell death”. Which cell death is not regulated? If cell death is controlled, then it should be beneficial.

4) The sentence in lines 248-253 is too long and needs to be broken into multiple sentences.

5) In lines 191-193, the authors included data regarding the clinical strain of bacteria, but there is no context. The authors should emphasize that the observed phenotype also occurs during infection with pathogenic E.coli. Information about the clinical isolate should be provided. I could not find the information in figure legends and materials/methods. Figures 2A and 2B should be labeled with the corresponding bacterial strain used to infect mice.

6) The authors need to indicate how they performed lung histology scoring.

7) Statistical analysis in Figures 1B, 1C, 1D, and 1E should be performed by two-way ANOVA.

8) In Fig.S2H, the authors need to provide zoomed images of neutrophils from Casp1 KO mice to compare to WT neutrophils.

9) In line 241. No need to start a new sentence.

10) In lines 243-245, the authors stated, “These findings also imply that caspase-1 may not act directly in the upstream of MLKL in this context, suggesting a potential alternative pathway in regulating neutrophil necroptosis during infection”. I am not following this train of thought. What data supports this conclusion?

11) No need for a new sentence in line 279.

12) In Figure S4E, treating mice with Anakinra is protective. The authors only used 6 mice, which may be insufficient to perform statistical analysis. A power analysis to identify sample size is needed. Provide the actual p-value for this graph.

13) The end of the paragraph in line 286 needs to have a conclusion sentence.

14) Two-way ANOVA must perform statistical analysis for figures 3C, 3D, and 3E.

15) Two-way ANOVA must perform statistical analysis for figures 4A, 4F, and 4G.

16) No need for a new sentence in line 320.

17) Two-way ANOVA must perform statistical analysis for figures S5B and S5C.

18) The sentence in lines 330-333 is too long and needs to be broken into multiple sentences.

19) No need for a new sentence in lines 341 and 345.

20) There is an extra period at the end of the sentence in line 342.

21) The end of the paragraph in line 350 needs to have a conclusion sentence.

22) No need for a new sentence in line 358.

23) Weirdly, poly(I:C) does not label control neutrophils in Figure 5G. How long the cells were stimulated?

24) The end of the paragraph in line 370 needs to have a conclusion sentence.

25) Two-way ANOVA must perform statistical analysis for figures 5F and 5H.

26) In Figure 6D, the authors indicated that they used WT control littermates. This should be labeled in the figures. Replace WT with either Zbp1+/+ or Zbp1+/- which is used in the experiment.

27) If control littermates were used in figures as indicated in lines 464-466, the authors should replace "WT" to indicate the WT mice genotype (i.e., Casp1+/+ or Casp1+/-).

28) No need for a new sentence in lines 379 and 383.

29) Two-way ANOVA must perform statistical analysis for Figure 6E.

30) No need for a new sentence in lines 531 and 537.

31) No need for a new sentence in lines 558, 559, 620, and 642.

PLOS authors have the option to publish the peer review history of their article (what does this mean? ). If published, this will include your full peer review and any attached files.

**Do you want your identity to be public for this peer review?** For information about this choice, including consent withdrawal, please see our Privacy Policy .

Reviewer #1: No

Reviewer #3: No

**Figure resubmission:**
---

## [Decision Letter · Decision Letter 2]

3 Apr 2025

PPATHOGENS-D-24-00639R2

Caspase-1-licensed pyroptosis drives dsRNA-mediated necroptosis and dampens host defense against bacterial pneumonia

PLOS Pathogens

Dear Dr. Chen,

Thank you for submitting your manuscript to PLOS Pathogens. After careful consideration, we feel that it has merit but does not fully meet PLOS Pathogens's publication criteria as it currently stands. Therefore, we invite you to submit a revised version of the manuscript that addresses the points raised during the review process. At this point no further experiments are required, but we will ask you to address the Minor Comments and Major Comment #7 brought up by Reviewer 3 as these concerns relate to clarity issues and specific statistical tests that were used. 

Please submit your revised manuscript within 30 days Jun 02 2025 11:59PM. If you will need more time than this to complete your revisions, please reply to this message or contact the journal office at plospathogens@plos.org. Please include the following items when submitting your revised manuscript:

We look forward to receiving your revised manuscript.

Kind regards,

Andrew J Monteith, Ph.D.

Guest Editor

PLOS Pathogens

Matthew Wolfgang

Section Editor

PLOS Pathogens

Sumita Bhaduri-McIntosh

Editor-in-Chief

PLOS Pathogens

orcid.org/0000-0003-2946-9497

Michael Malim

Editor-in-Chief

PLOS Pathogens

orcid.org/0000-0002-7699-2064

**Reviewers' Comments:**

Reviewer's Responses to Questions

**Part I - Summary**

Reviewer #1: The authors have made substantial revisions, which markedly improve an already strong manuscript. The work provides an important characterization of non-cell autonomous crosstalk between cell death pathways in a model of bacterial pathogenesis. The major findings are well supported by the data and meaningfully advance the field.

Reviewer #3: Although the authors did a somewhat good job incorporating the new data in the manuscript per previous review comments, several major concerns, especially technical concerns about the immunofluorescence microscopy images, need to be addressed. See below:

**Part II – Major Issues: Key Experiments Required for Acceptance**

Reviewer #1: (No Response)

Reviewer #3: 1) The immunofluorescence staining in Fig. 5G is not convincing to be specific. Why there is only one dot of FITC-poly(I: C) on neutrophils while fluorescent-positive cells can be easily directed by flow cytometry in Fig. 5F. The staining needs to be validated by using other cells like splenocytes or macrophages. The purpose of using PKH26 is to label cell boundaries, but clearly, this dye does not work for neutrophils. Why do DAPI-stained control cells not show classical neutrophil staining (polymorphonuclear)? The authors need to show images of many cells (lower magnification and zoomed). The provided images also appear blurry.

2) CD14-positive cells are not shown in Fig. S3B. The authors should use different areas of the slide where you can find the monocytes. The zoomed images of Casp1 KO mice are missing.

3) The Ly6G staining in Fig. S4 does not seem to be specific. The cells don’t look like neutrophils but epithelial cells. The staining looks different than the images from Fig. S3B, which represent stained neutrophils.

4) The bacterial burdens in the lungs seem to be too high. Mice were inoculated with 10^6 CFUs and yet there were 10^8 CFUs recovered after 12h. This is a two-log increase in bacterial counts in vivo within 12h, which seems abnormal. The authors need to check their calculations.

5) To compare the bactericidal function of Caspase 1 deficient neutrophils to wild-type neutrophils as stated in line 157, It is best to directly measure bactericidal activities.

6) It is not clear how WT and Casp1 KO neutrophils are dying from E.coli infection in Fig. S6A. What is the MOI and how long the infection was carried out? Do the neutrophils die due to bacterial overgrowth? If so, a lower MOI must be used. This needs some clarification and also needs to be mentioned in the main text.

7) Histology scoring criteria were based on the severity of damaged lungs. How the damage was quantified remained unclear. The word “damage” is ambiguous for assessing disease severity.

8) The authors’ conclusion that “the neutrophils from these mice did not express TLR3” needs rigorous assessment. Neutrophils from wild-type mice should express TLR3. Neutrophils response to TLR3 agonist in Fig. 6F and Fig. 5F, which implies that it is expressed. The RNAseq in Fig. 5A showed relative expression of the gene and was not simply present or absent. The data was also performed from n=2, which is insufficient to perform statistics. The immunoblot in Fig. S9 showed there is less TLR3 in Casp1 KO splenocytes compared to the wild-type. The authors need to confirm the specificity and sensitivity of the antibody that they used.

**Part III – Minor Issues: Editorial and Data Presentation Modifications**

Reviewer #1: (No Response)

Reviewer #3: 1) Statements in lines 81-85 must be referenced.

2) The statement in lines 152-154 “Although we observed more tissue-resident macrophages in the Casp1-/- mice in the physiological state, these cells almost completely disappeared at 12 hours after E. coli infection” is incorrect. Fig. 1D showed 10^4 macrophages at 12h, They have not completely disappeared as stated. Also, what did the authors mean by recruited macrophages? Macrophages are not recruited, monocytes do and then differentiate into macophages. Provide information in the figure legend on what markers were used to discriminate between different immune cells.

3) It is not clear how the RNAseq in Fig. S1J shows similar antimicrobial defense signals were induced by the two mouse strains (WT and Casp1 KO). There is only one set of data. What is the n=2 represent in there? How many mice per group were used for the RNA sequencing?

4) What cells are referred to in line 165?

5) Need a figure caption to the findings in lines 206-212.

6) For Fig. 2D, the authors must provide quantifications for all the bands, it is not clear why they chose to quantify only selected bands. If other bands are unnecessary, then another blot should be provided.

7) Change the double dose of bacteria in Fig. 3B to actual CFUs

8) Change “depletion” in line 254 to deletion.

9) What is L929+T/Z in Fig. 4D? Please explain why this is done in the main text and needs to be mentioned in the figure legend.

10) Fig. 4F needs a better label to indicate that neutrophils were incubated with supernatants from indicated epithelial cell lines that were left uninfected (PBS) or infected (E.coli).

11) Fig. S8E was mentioned after Fig. S8F to H.

12) The authors concluded that administration of dsRNA antibody during E. coli infection strongly protected the mice in lines 375-376. However, the data (Fig. 5I) showed that the administration of dsRNA antibodies weakly protected mice from infection.

13) There is no panel H in Fig. S2 as stated in the figure legend.

14) The quantification in Fig. S3B must be performed by One-way ANOVA.

15) The histology images in Fig. S5D, Fig. S8D, and Fig. S8E need to be labeled

16) Statistical analysis in Fig. S5 A, B, and C must be performed using Two-way ANOVA.

17) What are the histograms shown in Fig. 7A? The x-axis needs to be labeled.

18) Statistical analysis in Fig. S7D must be performed using Owo-way ANOVA. All statistically significant data needs to be marked.

19) There should not be any statistical test performed on the image in Fig. 9A. Correct the figure legend.

PLOS authors have the option to publish the peer review history of their article (what does this mean? ). If published, this will include your full peer review and any attached files.

**Do you want your identity to be public for this peer review?** For information about this choice, including consent withdrawal, please see our Privacy Policy .

Reviewer #1: No

Reviewer #3: No

**Figure resubmission:**
---

## [Editor Report · Decision Letter 3]

29 Apr 2025

Dear Professor Chen,

We are pleased to inform you that your manuscript 'Caspase-1-licensed pyroptosis drives dsRNA-mediated necroptosis and dampens host defense against bacterial pneumonia' has been provisionally accepted for publication in PLOS Pathogens.

Best regards,

Andrew J Monteith, Ph.D.

Guest Editor

PLOS Pathogens

Matthew Wolfgang

Section Editor

PLOS Pathogens

Sumita Bhaduri-McIntosh

Editor-in-Chief

PLOS Pathogens

orcid.org/0000-0003-2946-9497

Michael Malim

Editor-in-Chief

PLOS Pathogens

orcid.org/0000-0002-7699-2064
---

## [Editor Report · Acceptance letter]

Dear Professor Chen,

We are delighted to inform you that your manuscript, "Caspase-1-licensed pyroptosis drives dsRNA-mediated necroptosis and dampens host defense against bacterial pneumonia," has been formally accepted for publication in PLOS Pathogens.

Best regards,

Sumita Bhaduri-McIntosh

Editor-in-Chief

PLOS Pathogens

orcid.org/0000-0003-2946-9497

Michael Malim

Editor-in-Chief

PLOS Pathogens

orcid.org/0000-0002-7699-2064